# APPROXIMATING INSTANCE-DEPENDENT NOISE VIA INSTANCE-CONFIDENCE EMBEDDING

## ABSTRACT

Label noise in multiclass classification is a major obstacle to the deployment of learning systems. However, unlike the widely used *class-conditional noise* (CCN) assumption that the noisy label is independent of the input feature given the true label, label noise in real-world datasets can be aleatory and heavily dependent on individual instances. In this work, we investigate the *instance-dependent noise* (IDN) model and propose an efficient approximation of IDN to capture the instance-specific label corruption. Concretely, noting the fact that most columns of the IDN transition matrix have only limited influence on the class-posterior estimation, we propose a variational approximation that uses a single-scalar *confidence* parameter. To cope with the situation where the mapping from the instance to its confidence value could vary significantly for two adjacent instances, we suggest using *instance embedding* that assigns a trainable parameter to each instance. The resulting *instance-confidence embedding* (ICE) method not only performs well under label noise but also can effectively detect ambiguous or mislabeled instances. We validate its utility on various image and text classification tasks.

## 1 INTRODUCTION

In modern machine learning, large-scale data has become indispensable (Russakovsky et al., 2015; Wang et al., 2019a). A prevalent approach to collecting large-scale labeled datasets is to use imperfect sources such as crowdsourcing and web crawling (Fergus et al., 2005; Schroff et al., 2010; Wang et al., 2019a), which is usually less expensive and time-consuming than manual annotation by domain experts. However, such methods inevitably introduce label noise that may lead to overfitting and hurt the generalization of deep models (Arpit et al., 2017; Zhang et al., 2017).

In such situations, it is often beneficial to (i) remove mislabeled data or abstain from using confusing instances (Hara et al., 2019; Thulasidasan et al., 2019); (ii) increase robustness and reduce harmful influences of noisy labels (Malach & Shalev-Shwartz, 2017; Mirzasoleiman et al., 2020; Liu et al., 2020); or (iii) explicitly model the transition from the unobservable true label to the noisy observation (Goldberger & Ben-Reuven, 2017; Patrini et al., 2017; Xia et al., 2020). In this work, we focus on explicit modeling of the label corruption process, which is model-agnostic and data-efficient.

Most existing studies in this direction employ the *class-conditional noise* (CCN) assumption, i.e., the noisy label is independent of the input feature given the true label (Angluin & Laird, 1988; Natarajan et al., 2013; Patrini et al., 2017). However, this assumption could be too strong to fit some real-world data well (Xiao et al., 2015; Chen et al., 2021; Liu, 2021). More importantly, CCN only captures the general label flipping patterns between classes for all instances. In applications such as data cleansing and human-in-the-loop interaction, instance-specific noise information itself could be of central interest. This urges us to consider not only the class-conditional noise pattern but also the instance-specific noise modeling.

To handle this problem, in this work, we study the *instance-dependent noise* (IDN) model, where the noisy label also depends on the input. Several methods have been reported in the literature, but they either only focus on binary classification under strong assumptions (Menon et al., 2018; Cheng et al., 2020), are based on domain-specific knowledge (Xia et al., 2020), or need extra supervision (Berthon et al., 2021). In contrast, we propose a simple domain-agnostic approximation method for the multiclass IDN model, referred to as *instance-confidence embedding* (ICE). Concretely, to avoid estimating the noise transition matrix for each instance, we propose a variational approximation that

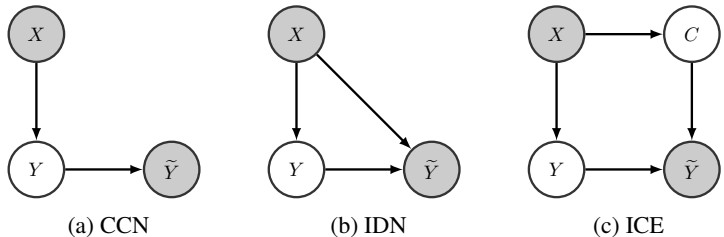

(a) CCN                    (b) IDN                    (c) ICE

Figure 1: **Graphical representations** of noise models, including the class-conditional noise (CCN) model, the instance-dependent noise (IDN) model, and the proposed *instance-confidence embedding* (ICE) approximation of IDN. Here, $X$ is the *input feature*, $Y$ is the *true label*, $\widetilde{Y}$ is the *noisy label*, and $C \in [0, 1]$ is a scalar *confidence* parameter.

uses a scalar *confidence* parameter (Section 3.2). Then, we suggest using *instance embedding* that assigns a trainable parameter to each instance because the mapping from the instance to its confidence value could be non-smooth and is usually not required to generalize to unseen examples (Section 3.3). Lastly, we show the effectiveness of the proposed method and its ability to detect ambiguous or mislabeled instances through experiments on various image and text classification tasks (Section 5).

## 2 PROBLEM: INSTANCE-DEPENDENT NOISE

In this section, we give a brief overview of learning with *instance-dependent noise* (IDN).

### 2.1 NOTATION

Consider a $K$-class classification problem, where $X \in \mathcal{X}$ is the *input feature* and $Y \in \{1, \ldots, K\}$ is the unobservable categorical *true label*. We assume that the *clean class-posterior* $p(Y|X)$ comes from a parametric family of distributions:

$$p_\phi(Y|X) := \text{Categorical}(Y|\boldsymbol{p} = f(X; \phi)), \tag{1}$$

where $\boldsymbol{p} \in \Delta^{K-1}$ is the probability parameter for $Y$ in the $(K-1)$-dimensional probability simplex $\Delta^{K-1}$, and $f : \mathcal{X} \to \Delta^{K-1}$ is a differentiable function parameterized by $\phi$ that maps the feature $X$ to its corresponding probability parameter $\boldsymbol{p}$. Then, let $\widetilde{Y} \in \{1, \ldots, K\}$ be the *noisy label*. The goal is to predict $Y$ from $X$ based on a finite i.i.d. sample of $(X, \widetilde{Y})$-pairs.

### 2.2 DEPENDENCE

Next, we introduce the dependence structure between $X$, $Y$, and $\widetilde{Y}$, which characterize different noise models. The graphical representations of noise models are illustrated in Fig. 1.

In IDN, we assume that the joint distribution of $X$, $Y$, and $\widetilde{Y}$ can be factorized as follows:

$$p(X, Y, \widetilde{Y}) = p(\widetilde{Y}|Y, X)p_\phi(Y|X)p(X). \tag{2}$$

That is, the noisy label $\widetilde{Y}$ depends on both the instance $X$ and the true label $Y$. Then, the *noisy class-posterior* $p(\widetilde{Y}|X)$ can be obtained by marginalizing $p(Y, \widetilde{Y}|X)$ over $Y$:

$$p_\phi(\widetilde{Y}|X) := \text{Categorical}(\widetilde{Y}|\boldsymbol{q} = \textstyle\sum_{Y=1}^{K} p(\widetilde{Y}|Y, X)p_\phi(Y|X)), \tag{3}$$

where $\boldsymbol{q} \in \Delta^{K-1}$ denotes the probability parameter for $\widetilde{Y}$.

Note that $p(\widetilde{Y}|Y, X)$ plays a central role in IDN. Since both $Y$ and $\widetilde{Y}$ are categorical random variables, for a certain instance $x$, $p(\widetilde{Y}|Y, X = x)$ can be seen as a $K \times K$ stochastic matrix $\boldsymbol{T}(x)$, whose elements are $\boldsymbol{T}_{ij}(x) := p(\widetilde{Y} = j|Y = i, X = x)$ for $i, j \in \{1, \ldots, K\}$. Conventionally, $\boldsymbol{T}(x)$ is called a *noise transition matrix* (Patrini et al., 2017). Then, $p(\widetilde{Y}|Y, X)$ can be regarded as a matrix-valued function $\boldsymbol{T} : \mathcal{X} \to [0, 1]^{K \times K}$ that maps each instance $x$ to its corresponding IDN transition matrix $\boldsymbol{T}(x)$. Without any restriction, we need $K \times K$ parameters for each instance $x$.

## 2.3 APPROACH

Owing to its complexity, IDN has only been studied to a limited extent but is of great interest recently. A straightforward method is to jointly estimate the matrix-valued function $\boldsymbol{T}(x)$ as well as the clean class-posterior $p_\phi(Y|X)$ using neural networks (Goldberger & Ben-Reuven, 2017). However, the estimation error of $\boldsymbol{T}(x)$ could be high, which deteriorates the classification performance. Another direction is to restrict the problem under certain conditions, so that we can provide theoretical guarantees (Menon et al., 2018; Cheng et al., 2020). However, existing work mainly focused on binary classification.

A promising approach is to approximate IDN using a simpler dependence structure, such as a mixture of noises with different semantic meanings (Xiao et al., 2015) or a weighted combination of noises that depend on parts of the instance (Xia et al., 2020). In this work, we also suggest that it might be unnecessary to obtain a $K \times K$ matrix for each instance $x$: Note that $p_\phi(\widetilde{Y}|x)$ can be seen as a linear combination of columns of $\boldsymbol{T}(x)$ weighted by $p_\phi(Y|x)$; If the maximum value of $p_\phi(Y|x)$ is close to 1, i.e., the label of the instance $x$ is almost deterministic, the estimation of $K-1$ columns of $\boldsymbol{T}(x)$ has only limited influence on the estimated noisy class-posterior $\widehat{p}(\widetilde{Y}|x)$. This suggests the possibility of using a relatively simple model to approximate $p(\widetilde{Y}|X)$ in real-world applications. In this work, we consider a *single-parameter* approximation for each instance, which is introduced in Section 3.2 and illustrated in Fig. 2.

Another issue is that existing methods still introduce some level of smoothness w.r.t. $x$ into $\boldsymbol{T}(x)$ (Goldberger & Ben-Reuven, 2017; Xiao et al., 2015; Xia et al., 2020). In real-world problems, however, we can only access a finite sample of $(X, \widetilde{Y})$-pairs that are possibly annotated by non-experts or web crawlers (Fergus et al., 2005). Thus, the label noise could be aleatory and $\boldsymbol{T}(x)$ could vary significantly for two adjacent instances. Also, the classifier $p_\phi(Y|X)$ is desired but the generalization of $\boldsymbol{T}(x)$ to unseen examples is usually dispensable. This inspires us to use *instance embedding* instead of neural network approximation, which is discussed in Section 3.3 and demonstrated in Fig. 3.

# 3 PROPOSED METHOD

In this section, we present our proposed method, *instance-confidence embedding* (ICE).

## 3.1 VARIATIONAL LOWER BOUND

Note that $\boldsymbol{T}(x)$ serves as a *linear mapping* from $\boldsymbol{p}$ to $\boldsymbol{q}$ (Eq. (3)). Due to the difficulty of estimating the matrix-valued function $\boldsymbol{T}(x)$, we use a simpler function $q_{\theta,\phi}(\widetilde{Y}|X)$ parameterized by $\theta$ as a *variational approximation* to $p_\phi(\widetilde{Y}|X)$. The choice of the approximation family is discussed in Section 3.2.

Then, let us consider the *expected log-likelihood* as the learning objective, which can be rewritten as

$$\underset{\widetilde{Y} \sim p(\widetilde{Y}|X)}{\mathbb{E}} [\log p(\widetilde{Y}|X)] = D_{\mathrm{KL}}(p_\phi(\widetilde{Y}|X) \parallel q_{\theta,\phi}(\widetilde{Y}|X)) + \mathcal{L}(\theta, \phi; X), \tag{4}$$

where $D_{\mathrm{KL}}$ denotes the Kullback-Leibler (KL) divergence, and the second term is

$$\mathcal{L}(\theta, \phi; X) := \underset{\widetilde{Y} \sim p(\widetilde{Y}|X)}{\mathbb{E}} [\log q_{\theta,\phi}(\widetilde{Y}|X)]. \tag{5}$$

Since the KL-divergence is always non-negative, this term gives a *variational lower bound* of the expected log-likelihood. Then, we have the following learning objective to maximize:

$$L(\theta, \phi) := \underset{X \sim p(X)}{\mathbb{E}} [\mathcal{L}(\theta, \phi; X)] = \underset{X, \widetilde{Y} \sim p(X, \widetilde{Y})}{\mathbb{E}} [\log q_{\theta,\phi}(\widetilde{Y}|X)]. \tag{6}$$

In practice, the expectation can be approximated using the empirical distribution based on a finite i.i.d. sample of $(X, \widetilde{Y})$-pairs.

| Clean | Linear interpolation | Power transformation | Noisy |
|---|---|---|---|
| 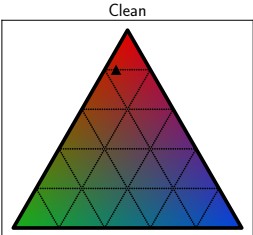 | 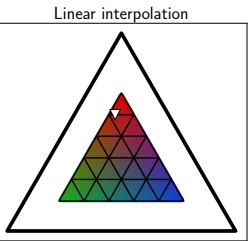 | 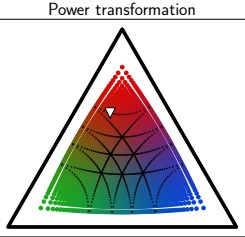 | 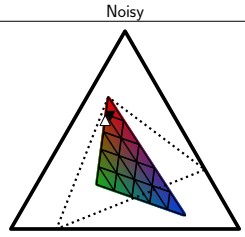 |

Figure 2: An illustration of the transformation ($\blacktriangle \mapsto \blacktriangledown$) from the clean class-posterior $p_\phi(Y|x)$ (the leftmost) to the noisy class-posterior $p_\phi(\widetilde{Y}|x)$ (the rightmost). The outer black triangle depicts the probability simplex $\Delta^2$ projected to the 2-dimensional space for 3-class classification. We can see that when the label is almost deterministic ($\blacktriangle$ is close to a vertex), the estimation of $K-1$ columns of the transition matrix $\boldsymbol{T}(x)$ (the two deviated vertices of the dotted triangle) has only limited influence on the estimated noisy class-posterior $\widehat{p}(\widetilde{Y}|x)$ ($\triangle$ is still close to $\blacktriangledown$). This inspires us to go a step further and use **single-parameter approximations** ($\blacktriangle \mapsto \triangledown$) $q_{\theta,\phi}(\widetilde{Y}|x)$ (Eqs. (8) and (9)).

## 3.2 VARIATIONAL APPROXIMATION

Next, we discuss the choice of the variational approximation family of $q_{\theta,\phi}(\widetilde{Y}|X)$.

To approximate the effect of multiplying an IDN transition matrix $\boldsymbol{T}(x)$ that requires $K \times K$ parameters for each instance $x$, in this work, we use a simpler transformation from $\boldsymbol{p}$ to $\boldsymbol{q}$, which is not necessarily linear. Compared with estimating a full matrix for each instance without any restriction (Goldberger & Ben-Reuven, 2017), obtaining only an approximation may cause higher approximation error, but on the other hand, it may reduce estimation error and thus improve the classification performance. The high estimation error of complex models might be more harmful, which is empirically validated in Section 5. It is also the case when using CCN as an approximation of IDN to balance this trade-off. The difference is that CCN obtains a *complete* transition matrix common to *all* instances, but ICE aims to obtain an *approximated* trend for *each* instance, which gives useful instance-specific noise information.

Then, we suggest to use a *single-scalar parameter* $C \in [0, 1]$ for each instance to control this approximation, which is useful for sorting and comparing training examples. This design might also be useful in data cleansing, learning with rejection, or active learning (Hara et al., 2019; Thulasidasan et al., 2019; Charoenphakdee et al., 2021). This parameter is referred to as the *confidence* and is obtained via a function $g : \mathcal{X} \to [0, 1]$ parameterized by $\theta$, i.e., $C = g(X; \theta)$. Then, $\theta$ can be regarded as the collection of $C$ for all instances. The confidence $C$ plays a central role in our method, where $C = 0$ means that the instance is ambiguous or mislabeled and thus the classifier should not give a confident prediction.

Finally, we need to design a transformation from $\boldsymbol{p}$ to $\boldsymbol{q}$ parameterized by the confidence $C$. We denote this function by $h : \Delta^{K-1} \to \Delta^{K-1}$. In summary, $q_{\phi,\theta}(\widetilde{Y}|X)$ takes the following form:

$$q_{\phi,\theta}(\widetilde{Y}|X) := \text{Categorical}(\widetilde{Y}|\boldsymbol{q} = h(f(X; \phi); g(X; \theta))). \tag{7}$$

Next, we analyze what characteristics $h$ needs to have. First, we suggest that $h$ needs not necessarily to be a linear transformation because the transformation is instance-dependent and any function that maps $\boldsymbol{p}$ to $\boldsymbol{q}$ as close as possible for a certain instance $x$ would suffice. Second, we require that $\arg\max(\boldsymbol{p}) = \arg\max(\boldsymbol{q})$, i.e., $\boldsymbol{q} = h(\boldsymbol{p}; C)$ should be an *argmax-preserving* function so that the top-1 index of the probability vector does not change. This is because $h$ should only affect the confidence of the prediction, not the final decision. Otherwise, if $h$ is too flexible and is able to map a confident prediction to a different confident prediction, then the output of $f$ could be arbitrary, and consequently, no information of the true label can be learned from the noisy label supervision.

Based on this motivation and the aforementioned semantics of $C$, we require that $h(\boldsymbol{p}; 1) = \boldsymbol{p}$ and $h(\boldsymbol{p}; 0) = \boldsymbol{u}$, where $\boldsymbol{u} \in \Delta^{K-1}$ is the uniform probability vector ($\boldsymbol{u}_i = \frac{1}{K}$). Then, when the confidence $C$ is high, the classifier gives a prediction closer to the original *confident prediction* $\boldsymbol{p}$; and when the confidence $C$ is low, the classifier tends to give a *random guess* $\boldsymbol{u}$.

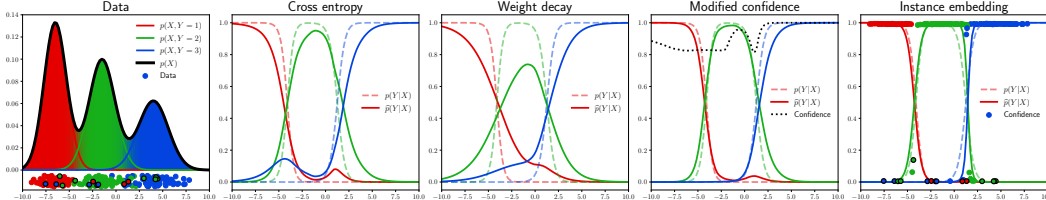

Figure 3: An example of the learned **class-posteriors** using (a) the usual cross entropy without any modification; (b) weight decay; (c) modified confidence of the prediction (Eq. (8)) with neural network approximation for $g : \mathcal{X} \rightarrow [0, 1]$; and (d) instance-confidence embedding. The points with black edges are mislabeled instances. We can observe that noisy labels affect the decision boundary and the model complexity. Comparing the last two panels, modifying the confidence of the prediction works better with instance embedding than neural network approximation. ICE can reduce the influence of ambiguous or mislabeled instances to improve the class-posterior estimation.

Here, we propose two functions for $h$ that satisfy the above conditions:

$$\boldsymbol{q}_i = C\boldsymbol{p}_i + (1 - C)\boldsymbol{u}_i, \qquad \text{(linear interpolation)} \qquad (8)$$

$$\boldsymbol{q}_i = \frac{\boldsymbol{p}_i^C}{\sum_{j=1}^{K} \boldsymbol{p}_j^C} \propto \boldsymbol{p}_i^C, \qquad \text{(power transformation)} \qquad (9)$$

for $i = 1, \ldots, K$. The visualization of these two transformations for $K = 3$ is given in Fig. 2.

### 3.3 INSTANCE EMBEDDING

The last piece of our method is the choice of $g : \mathcal{X} \rightarrow [0, 1]$, the function that maps the instance $x$ to its confidence value $C$. It is also possible to use a neural network to approximate this function. However, because we usually only have a limited number of training examples and $g$ could be non-smooth w.r.t. its input $x$, $g$ may not be well approximated by a neural network with similar complexity to the classifier $f$, which is illustrated in an example in Fig. 3. Further, $g$ may be rarely needed after training so its generalization ability is not required in many cases.

Based on these facts, we propose to use *instance embedding*, i.e., to assign a trainable parameter to each instance $x$. In other words, the only feature for an instance we use is its *index* in the training dataset. In this way, $g$ is expressive and flexible but cannot be used for predicting the confidence of unseen instances. Accordingly, for a training dataset of size $N$, we need $N$ parameters for a one-dimensional instance embedding.

This seems to be a high additional computational cost when the dataset size is large, but it is often acceptable, because (i) in modern deep learning, it is common to use over-parameterized models (Nakkiran et al., 2019), and the number of instances is usually not as large as the number of parameters of the classifier $f$ (e.g., CIFAR-10 (Krizhevsky, 2009): $5 \times 10^4$, ResNet-18 (He et al., 2016): $\sim 1 \times 10^7$); and (ii) the gradient of the instance embedding is sparse and only a small subset of parameters needs to be updated at each iteration.

The idea of associating an entity with a scalar or vector embedding using a simple lookup table with a fixed dictionary size has been widely used in natural language processing (Mikolov et al., 2013; Pennington et al., 2014; Peters et al., 2018; Devlin et al., 2019) due to the discrete nature of tokens, and can be seen recently in contrastive learning (Wu et al., 2018; He et al., 2020) for vision tasks. Instance embedding enables the function to take any possible value on all observed instances but cannot generalize to any unseen token or image.

Combining the components introduced above, the learning objective Eq. (6) is in the following form:

$$L(\theta, \phi) = \text{cross-entropy}\big(h(\overbrace{\underbrace{f(X; \phi)}_{\boldsymbol{p}, \text{ neural network classifier, Eq. (1)}} ; \underbrace{g(X; \theta)}_{C, \text{ instance-confidence embedding}}}^{\boldsymbol{q}, \text{ fixed transformation, Eqs. (8) and (9)}}), \widetilde{Y}\big).$$

We also provide code snippets in Appendix A for practitioners and gradient analysis in Appendix B for theoretical implications.

## 4  RELATED WORK

In this section, we review related problem settings and methods.

**Class-conditional noise (CCN).**  Compared with the IDN model, the instance-independent and *class-conditional noise* (CCN) model has an additional assumption: $p(\widetilde{Y}|Y, X) = p(\widetilde{Y}|Y)$, i.e., the noisy label $\widetilde{Y}$ only depends on the true label $Y$. CCN has been well studied in both binary (Angluin & Laird, 1988; Long & Servedio, 2010; Natarajan et al., 2013; Van Rooyen et al., 2015; Liu & Tao, 2015) and multiclass (Patrini et al., 2017; Xia et al., 2019; Yao et al., 2020; Zhang et al., 2021b; Zhu et al., 2021) classification. Also, *robust loss functions* (Ghosh et al., 2017; Zhang & Sabuncu, 2018; Wang et al., 2019b; Charoenphakdee et al., 2019; Ma et al., 2020; Feng et al., 2020; Lyu & Tsang, 2020; Liu & Guo, 2020) have been mainly developed under the CCN setting. In practice, CCN methods can serve as practical approximations of IDN but the assumption could be too strong to fit some real-world data well (Xiao et al., 2015).

**Label smoothing.**  Note that Eq. (8) is similar to the *label smoothing* (LS) technique (Szegedy et al., 2016; Pereyra et al., 2017; Lukasik et al., 2020), where the empirical distribution is linearly interpolated with a uniform distribution with a fixed mixing parameter. It is also related to the soft/hard *bootstrapping loss* (Reed et al., 2015), where the observed label is mixed with the predicted probability/predicted label. In contrast, in our method, it is the prediction $p$ that is "smoothed", not the label. We elucidate their relations and differences in Appendix C.

**Temperature scaling.**  If we use softmax as the final layer of the neural network for $p_\phi(Y|X)$, the proposed method is closely related to the *temperature scaling* (TS) technique (Guo et al., 2017). Concretely, if $p_i \propto \exp\{f_i(X; \phi)\}$ for $i = 1, \ldots, K$, then Eq. (9) becomes

$$q_i \propto \exp\{C f_i(X; \phi)\}, \tag{10}$$

which shows that $C$ is the *reciprocal of the temperature*. The difference is that the parameter $C$ is instance-dependent in our formulation, rather than being fixed for all instances. Also, TS (Guo et al., 2017) and its extensions (Kull et al., 2019; Rahimi et al., 2020) have been mainly used as post-hoc confidence calibration methods, while our method is used during training.

**Sample selection.**  In a broader sense, the proposed method belongs to a category of methods that treat training examples differently in order to reduce the harmful effects of mislabeled instances. Besides the class-posteriors that our method uses, these methods exploit the training dynamics, loss characteristics, gradient information, or information of data itself from various perspectives. Examples include *data cleansing* (Liu et al., 2008; Northcutt et al., 2019; Hara et al., 2019) that first removes harmful instances and then (re-)trains the model on the remaining subset; *dynamic training sample selection* (Malach & Shalev-Shwartz, 2017; Jiang et al., 2018; Han et al., 2018; Wang et al., 2018; Yu et al., 2019; Mirzasoleiman et al., 2020; Wu et al., 2020; Chen et al., 2021; Cheng et al., 2021; Zhang et al., 2021a) that selects training examples dynamically during training; *training techniques* (Menon et al., 2020; Liu et al., 2020) that are designed to increase robustness and avoid memorization of noisy labels; *learning with rejection* or *selective classification* (El-Yaniv & Wiener, 2010; Thulasidasan et al., 2019; Mozannar & Sontag, 2020; Charoenphakdee et al., 2021) that abstains from using confusing instances while improving the classification performance on accepted instances; and *semi-supervised learning* (Nguyen et al., 2020; Li et al., 2020) that exploits unlabeled data. We discuss the relationship with some of them in more detail in Appendix D.

In the same spirit, our proposed method also attempts to detect harmful instances and reduce their influences automatically so as to improve the robustness of the class-posterior estimation. However, unlike explicit sample selection methods, the resulting algorithm is lightweight and has a low computational cost. Also, because the proposed method only affects the class-posterior, it is usually compatible with other training methods. Thus, the proposed method can be used alone or integrated into an existing training pipeline to further improve the performance.

Table 1: **Accuracy** (%) on the MNIST, FMNIST, KMNIST, SVHN, CIFAR-10, and CIFAR-100 datasets with instance-dependent noise. "Mean (standard deviation)" for 10 trials are reported. Outperforming methods are highlighted in boldface using one-tailed t-tests with a significance level of 0.05.

|  | MNIST | FMNIST | KMNIST | SVHN | CIFAR-10 | CIFAR-100 |
|---|---|---|---|---|---|---|
| CCE | 95.26(0.50) | 85.41(0.56) | 82.64(0.59) | 76.93(1.86) | 74.72(1.70) | 51.90(0.38) |
| Bootstrapping | 97.40(0.22) | 87.22(0.40) | 85.69(0.79) | 79.60(1.93) | 78.70(0.74) | 52.60(0.50) |
| Adaptation | 94.94(0.49) | 84.91(0.42) | 81.23(1.74) | 69.97(3.41) | 74.71(0.84) | 39.18(1.51) |
| Forward | 95.47(0.45) | 85.78(0.55) | 83.71(0.86) | 75.17(4.21) | 75.08(0.87) | 52.12(0.50) |
| Dual-T | 97.05(0.40) | 86.54(0.49) | 84.91(0.86) | 79.49(1.84) | 81.10(0.47) | 52.77(0.48) |
| DAC | 95.78(0.29) | 86.19(0.50) | 83.21(1.05) | 79.39(2.33) | 74.95(0.85) | 52.17(0.33) |
| GCE | 97.69(0.18) | 87.10(0.54) | 86.50(1.04) | 79.64(1.89) | 81.65(0.51) | 55.75(0.30) |
| PLC | **97.87(0.17)** | 87.69(0.45) | 87.30(0.49) | **86.23(0.37)** | 82.51(0.34) | 55.91(0.41) |
| ICE-LIN | 97.39(0.25) | 87.07(0.29) | 87.08(0.50) | 77.16(2.19) | 82.07(0.39) | **56.04(0.34)** |
| ICE-POW | **97.94(0.18)** | **87.90(0.36)** | **87.73(0.62)** | 81.82(0.99) | **82.37(0.29)** | 55.07(0.48) |

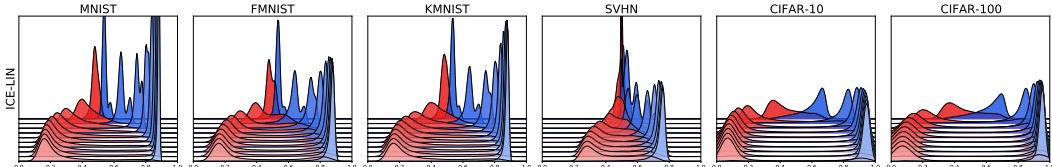

Figure 4: **Ridgeline plots** of the confidence $C$ during training. The red/blue curves represent the confidence of instances with flipped/original labels, respectively.

## 5 EXPERIMENTS

In this section, we experimentally verify if the proposed *instance-confidence embedding* (ICE) method is able to differentiate mislabeled instances from correct ones and consequently improve the classification performance. We also demonstrate that there already exist ambiguous or mislabeled training examples in the original datasets which can be detected by the proposed method.

### 5.1 IMAGE CLASSIFICATION

**Datasets.**  We evaluated our method on six image classification datasets, namely **MNIST** (Le-Cun et al., 1998), Fashion-MNIST (**FMNIST**) (Xiao et al., 2017), Kuzushiji-MNIST (**KMNIST**) (Clanuwat et al., 2018), **SVHN** (Netzer et al., 2011), **CIFAR-10**, and **CIFAR-100** (Krizhevsky, 2009) datasets. We used a method similar to the one used in Zhang et al. (2021a) to generate instance-dependent label noises whose overall noise rate is around 52%. See Appendix E for details and more experimental results.

**Methods.**  We compared the following ten methods: (1) (**CCE**) categorical cross-entropy loss; (2) (**Bootstrapping**) (hard) bootstrapping loss (Reed et al., 2015) that regularizes the output with the predicted label; (3) (**Adaptation**) noise adaptation layer (Goldberger & Ben-Reuven, 2017) that estimates a full $K \times K$ transition matrix for each instance; (4) (**Forward**) forward correction (Patrini et al., 2017) that estimates a transition matrix for all instances; (5) (**Dual-T**) dual-T estimator (Yao et al., 2020) that uses the normalized confusion matrix to correct the transition matrix; (6) (**DAC**) deep abstaining classifier (Thulasidasan et al., 2019) that uses abstention for robust learning; (7) (**GCE**) generalized cross-entropy loss (Zhang & Sabuncu, 2018) as a robust loss; (8) (**PLC**) progressive label correction (Zhang et al., 2021a); (9) (**ICE-LIN**) instance-confidence embedding with the linear interpolation (Eq. (8)); and (10) (**ICE-POW**) the one with the power transformation (Eq. (9)). For a fair comparison, we implemented aforementioned methods using the same network architecture and hyperparameters.

**Models.**  For MNIST, FMNIST, and KMNIST, we used a sequential convolutional neural network (CNN) and an Adam optimizer (Kingma & Ba, 2015). For SVHN, CIFAR-10 and CIFAR-100, we used a residual network model ResNet-18 (He et al., 2016) and a stochastic gradient descent (SGD) optimizer with momentum (Sutskever et al., 2013). Hyperparameters are provided in Appendix E.

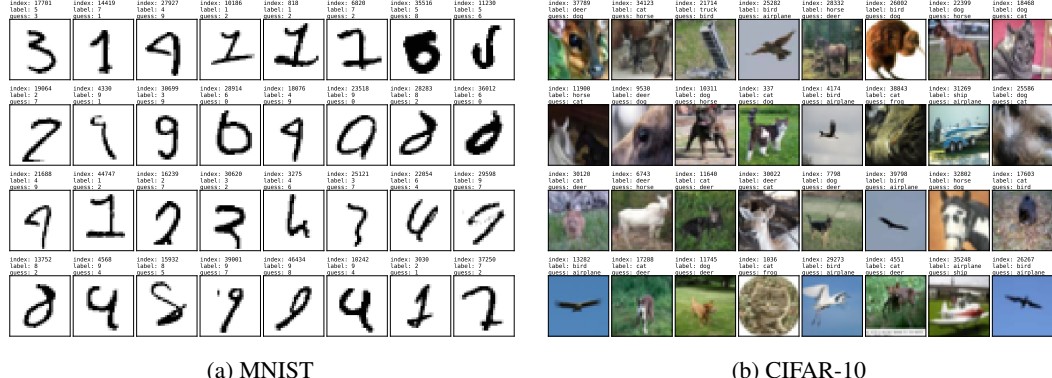

| (a) MNIST | (b) CIFAR-10 |

Figure 5: The 32 most **low-confidence training examples** in the MNIST and CIFAR-10 datasets, ordered left-right, top-down by increasing confidence. The index in the dataset, original label, and predicted label are annotated above each image for verification.

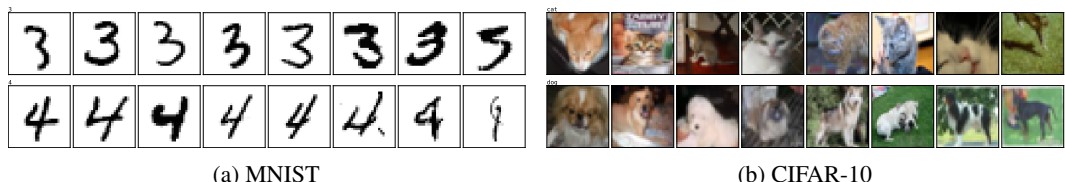

| (a) MNIST | (b) CIFAR-10 |

Figure 6: The **class-wise spectra** from high-confidence (left) to low-confidence (right) training examples in the MNIST (digit 3 and 4) and CIFAR-10 (cat and dog) datasets.

**Improving classification performance.** To verify if the proposed method is able to improve the classification performance under label noise, we constructed semi-synthetic noisy datasets so that the true labels are known. We regarded the original labels as clean labels, although as will be shown in the next experiment, label errors already exist in the original datasets to some extent. We ran 10 trials and reported the means and standard deviations of the test accuracy in Table 1.

We can observe that the proposed method generally outperforms the baseline methods. It is worth noting that when there are a large number of classes (e.g., CIFAR-100), estimating a full transition matrix for each instance (Adaptation) may deteriorate drastically because it requires an additional $K \times K$ output and the estimation error could be high. On the contrary, the complexity of our single-parameter approximation does not increase as the number of classes increases. Additionally, in the ridgeline plots in Fig. 4, we can observe the separation of instances with flipped/original labels using the learned confidence $C$, which may explain the performance improvement.

**Detecting ambiguous/mislabeled instances.** Next, we demonstrate that the proposed method can be used for detecting ambiguous or possibly mislabeled instances. We trained the model on the original datasets with the proposed method. A benefit of using a single-parameter approximation is that it naturally derives an order of the training examples. We sorted the training examples via the confidence and showed the 32 most low-confidence ones in Fig. 5. We also extracted high-/low-confidence training examples for each classes as shown in Fig. 6. Results of other datasets are provided in Appendix E.

We can observe that, surprisingly, in these supposedly clean datasets, a number of instance might be mislabeled. In MNIST and SVHN, we found clearly mislabeled images. There are ambiguous images such as 2-7 and 4-9 pairs in MNIST and shirt/T-shirt/pullover/coat photos in Fashion-MNIST. In CIFAR-10, it is interesting that images in the animal category are more likely to have a low confidence. We conjecture that the *spurious correlation* between the object and the background color plays an important role. We also found multi-modality issues, e.g., kiwi, owl, and chicken are all labeled as bird but are not *visually prototypical* birds. This phenomenon suggests the possibility of using the proposed method for diagnosing label issues in large-scale datasets.

Table 2: **Performance** (%) on the GLUE benchmark for natural language understanding. We reported Matthews correlation coefficient on CoLA, F1 score/accuracy on MRPC and QQP, and accuracy otherwise. MNLI-(m/mm) denotes MultiNLI matched/mismatched, respectively.

|     | CoLA  | SST2  | MRPC        | QQP         | MNLI-(m/mm) | QNLI  | RTE   | WNLI  |
|-----|-------|-------|-------------|-------------|-------------|-------|-------|-------|
| CCE | 54.76 | 92.55 | 88.04/82.35 | 87.80/90.96 | 83.83/84.31 | 90.77 | 66.43 | 50.70 |
| ICE | 57.83 | 92.20 | 89.54/85.05 | 87.85/90.92 | 83.81/84.36 | 91.14 | 63.90 | 56.34 |

Table 3: Six selected **low-confidence training examples** in the CoLA dataset.

| Index | Label        | Guess        | Text                                |
|-------|--------------|--------------|-------------------------------------|
| 390   | acceptable   | unacceptable | He I often sees Mary.               |
| 7756  | acceptable   | unacceptable | That monkey is ate the banana.      |
| 8332  | acceptable   | unacceptable | I wanted Jimmy for to come with me. |
| 2801  | unacceptable | acceptable   | Paula hit the sticks.               |
| 2479  | unacceptable | acceptable   | Kelly buttered the bread with butter. |
| 6795  | unacceptable | acceptable   | Henry wanted to possibly marry Fanny. |

Further, in the class-wise spectra of training examples (Fig. 6), we can observe that the confidence values may capture the level of ambiguity of instances. This suggests the possibility of using the proposed method in other tasks such as data cleansing, learning with rejection, and active learning.

## 5.2 TEXT CLASSIFICATION

We discovered that noisy label issues also exist in text datasets. We conducted similar experiments on the GLUE benchmark (Wang et al., 2019a), which is a collection of datasets for natural language understanding. We trained a BERT-base model pretrained using a masked language modeling (MLM) objective (Devlin et al., 2019) with a default AdamW optimizer (Loshchilov & Hutter, 2017). The performance in terms of the suggested evaluation metric was reported in Table 2.

We can observe that, except on the RTE dataset, the performance was improved or approximately the same compared with the default CCE method, which shows the benefits of using instance-specifically adjusted confidences. Although, if the dataset is relatively clean, the improvement might be marginal.

We also found mislabeled or ambiguous instances in these datasets. A typical example is the Corpus of Linguistic Acceptability (CoLA) (Warstadt et al., 2019) dataset, which consists of English grammatical acceptability judgments. Six selected low-confidence training examples are given in Table 3. We found that several ungrammatical sentences were mislabeled as acceptable, and some *syntactically acceptable* sentences were labeled as unacceptable by annotators possibly because they have *semantic errors*. In this way, we may use the proposed method to probe if the model prediction is consistent with our intent. More results are provided in Appendix E.

## 6 CONCLUSION

We have introduced a novel variational approximation of the instance-dependent noise (IDN) model, referred to as *instance-confidence embedding* (ICE). Compared with existing methods based on the class-conditional noise (CCN) assumption, the proposed method is able to capture instance-specific noise information and consequently improve the classification performance. The use of the one-dimensional instance embedding naturally derives an order of training examples which can be used for detecting ambiguous or mislabeled instances. For future directions, it is interesting to explore its combination with other training techniques and its extensions in data cleansing, learning with rejection, or active learning.

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

## A IMPLEMENTATION

In this section, we provide some implementation details of the proposed method for practitioners. Here, we only extract and highlight the core part to explain our method.

First, thanks to the development of PyTorch (Paszke et al., 2019), the instance embedding part can be actually implemented easily using `torch.nn.Embedding`. This is also partially why we choose "embedding" as part of the name of the proposed method. The weight is initialized to be $0$ for all instances and the embedding is followed by a sigmoid layer so the confidence value is within $[0, 1]$.

```python
def embedding(size):
    """
    size: the number of training examples
    """
    embed = nn.Embedding(size, 1, sparse=True)
    embed.weight.data.fill_(0.) # initialization
    embed = nn.Sequential(embed, nn.Sigmoid())
    return embed
```

Second, assume that a neural network `model` is defined and the index `i` of each training example is also provided, the learning objective can be calculated as follows:

```python
def loss(i, x, y):
    """
    i: index [batch_size]
    x: input [batch_size, ...]
    y: label [batch_size, num_classes]
    """
    t = model(x) # Section 2.1 Eq. (1)
    c = embed(i) # Section 3.3 instance embedding
    s = transformation(t, c) # Section 3.2 Eqs. (8) or (9)
    l = cross_entropy(s, y) # Section 3.1 Eq. (6)
    return l
```

Here, the `transformation` function corresponds to the approximation family Eqs. (8) and (9). Please note the slight difference that `t` and `s` are the logits (output without softmax), not the probabilities for the purposes of easy implementation and numerical stability. We can choose it from the following functions:

```python
def linear_interpolation(t, c):
    # Section 3.2 Eq. (8)
    return log(c * softmax(t, dim=1) + (1 - c) * 1 / t.shape[1])
```

```python
def power_transformation(t, c):
    # Section 3.2 Eq. (9) and Section 4 (temperature scaling)
    return c * t
```

Finally, The parameters of both `embed` and `model` can be optimized simultaneously using any gradient-based method such as `torch.optim.SGD` and `torch.optim.Adam`.

# B  GRADIENT ANALYSIS

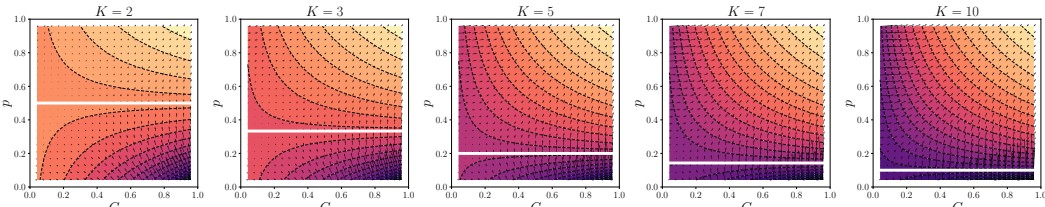

Figure 7: Contours of the log-likelihood w.r.t. $\boldsymbol{p}_i$ and $C$ using the linear interpolation (Eq. (8)) for $K$ in $\{2, 3, 5, 7, 10\}$.

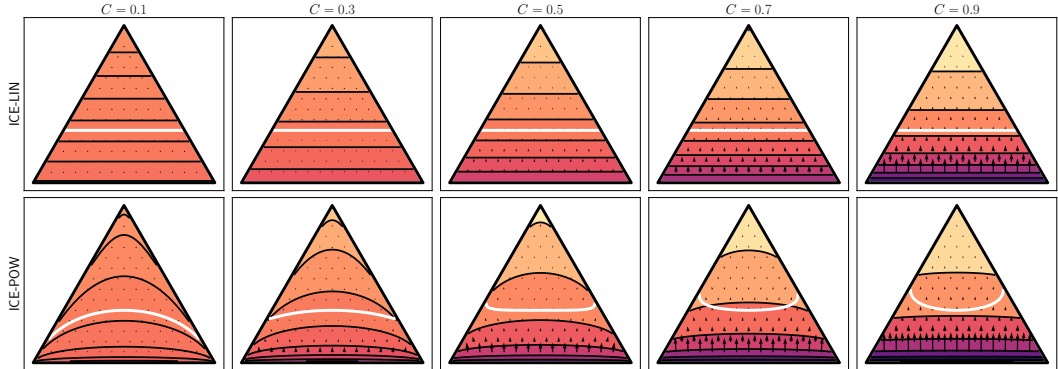

Figure 8: Contours of the log-likelihood on the simplex when $K = 3$ using the linear interpolation (Eq. (8), top) and the power transformation (Eq. (9), bottom) for $C$ in $\{0.1, 0.3, 0.5, 0.7, 0.9\}$.

In this section, we provide a basic gradient analysis and visualization for our proposed method.

The gradients of the log-likelihood using Eqs. (8) and (9) are

$$\frac{\partial}{\partial C} \log(\boldsymbol{q}_i) = \frac{\boldsymbol{p}_i - \frac{1}{K}}{C\boldsymbol{p}_i + (1 - C)\frac{1}{K}}, \qquad \text{(linear interpolation)} \qquad (11)$$

$$\frac{\partial}{\partial C} \log(\boldsymbol{q}_i) = \sum_{j=1}^{K} \boldsymbol{p}_j^C \log \frac{\boldsymbol{p}_i}{\boldsymbol{p}_j}, \qquad \text{(power transformation)} \qquad (12)$$

respectively. Their sign boundaries are $\boldsymbol{p}_i = \frac{1}{K}$ and $\boldsymbol{p}_i = e^{-H(\boldsymbol{q},\boldsymbol{p})}$, respectively, where $H(\cdot, \cdot)$ denotes the cross-entropy. We can find that for the linear interpolation (Eq. (8)), when $\boldsymbol{p}_{\widetilde{y}} < \frac{1}{K}$, $\frac{\partial L}{\partial C} < 0$ and when $\boldsymbol{p}_{\widetilde{y}} > \frac{1}{K}$, $\frac{\partial L}{\partial C} > 0$. Similarly, for the power transformation (Eq. (9)), when $\boldsymbol{p}_{\widetilde{y}} < e^{-H(\boldsymbol{q},\boldsymbol{p})}$, $\frac{\partial L}{\partial C} < 0$ and when $\boldsymbol{p}_{\widetilde{y}} > e^{-H(\boldsymbol{q},\boldsymbol{p})}$, $\frac{\partial L}{\partial C} > 0$.

The contours of the likelihood for different parameters are plotted in Figs. 7 and 8. Note that the class-posterior $\boldsymbol{p}$ is obtained from a neural network, so it can be influenced by other instances, especially adjacent instances. On the other hand, the confidence $C$ is obtained via instance embedding, so it can take any value independently. If the predicted class-posterior $\boldsymbol{p}_{\widetilde{y}}$ for an instance $x$ is low (e.g., because this instance is mislabeled and the majority of adjacent instances are predicted to belong to other classes), then the classifier tends to decrease its confidence value so as not to overfit this possibly mislabeled instance. The gradient magnitude is the largest when $C$ is high and $\boldsymbol{p}_{\widetilde{y}}$ is low (confident wrong prediction), the smallest when both $C$ and $\boldsymbol{p}_{\widetilde{y}}$ are high (confident correct prediction), and in the middle when $C$ is low (uncertain prediction like a random guess). In this way, we can equip the neural network model with an option of changing the confidence of prediction for individual training examples to mitigate overfitting possibly mislabeled instances.

## C   LINEAR INTERPOLATION

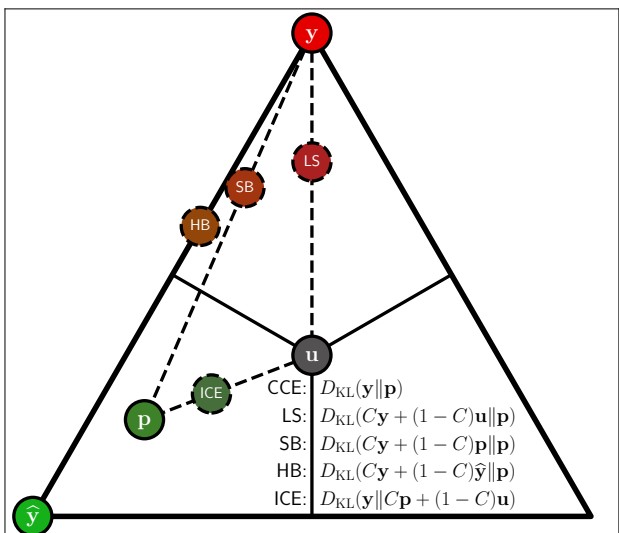

Figure 9: Illustration of related methods, including the categorical cross-entropy (CCE), label smoothing (LS), soft/hard bootstrapping loss (SB/HB), and the proposed instance-confidence embedding (ICE) with the linear transformation (Eq. (8)).

Linear interpolation between some properties of an instance and other value is a widely used technique for regularization in machine learning, such as the bootstrapping loss Reed et al. (2015) and the label smoothing technique (Szegedy et al., 2016; Pereyra et al., 2017; Lukasik et al., 2020). In this section, we briefly summarize related techniques and compare their differences.

Concretely, let $p$ be the predicted probability vector for $Y$ (Eq. (1)), $y$ be the one-hot vector for the observed label, $\widehat{y} = \arg\max p$ is the one-hot vector for the predicted label, $u$ be the uniform probability vector ($u_i = \dfrac{1}{K}$ for $i \in \{1, \ldots, K\}$). Here, $p, y, \widehat{y}, u \in \Delta^{K-1}$ are all in the probability simplex. Let $C \in [0, 1]$ be a scalar linear interpolation parameter.

Then, as also shown in Fig. 9, the learning objectives are equivalent to the following KL-divergences:

$$D_{\mathrm{KL}}(y \parallel p), \qquad \text{(categorical cross-entropy)} \tag{13}$$

$$D_{\mathrm{KL}}(Cy + (1 - C)u \parallel p), \qquad \text{(label smoothing)} \tag{14}$$

$$D_{\mathrm{KL}}(Cy + (1 - C)p \parallel p), \qquad \text{(soft bootstrapping loss)} \tag{15}$$

$$D_{\mathrm{KL}}(Cy + (1 - C)\widehat{y} \parallel p), \qquad \text{(hard bootstrapping loss)} \tag{16}$$

$$D_{\mathrm{KL}}(y \parallel Cp + (1 - C)u). \qquad \text{(instance-confidence embedding)} \tag{17}$$

We can see that the label smoothing and the bootstrapping loss methods smooth the target, but the proposed ICE method smooths the prediction. Note that it is impossible to let $C$ be an instance-dependent parameter in other methods, because when $C = 0$, the supervision signal $Y$ can be completely lost.

Another technique using linear interpolation is *mixup* (Zhang et al., 2018), which also interpolates the input features $X$ between two instances. Therefore its characteristics could be more different than the methods mentioned above.

## D  SAMPLE SELECTION

Our method can be regarded as a "soft" sample selection method that de-emphasizes the supervision from possibly mislabeled training examples to some extent. In the literature, several "hard" sample selection methods have been reported recently, e.g., Cheng et al. (2021); Zhang et al. (2021a). In this section, we discuss their relationship and the pros and cons of these methods.

**Soft vs. hard.**   Hard sample selection methods usually use a predefined thresholding rule based on the regularized loss [(Cheng et al., 2021), Eq. (4)] or the predicted probability [(Zhang et al., 2021a), Section 2.2] to select a subset of possibly correct training examples at each iteration. The threshold is either determined dynamically (Cheng et al., 2021) or decreased via a manually designed schedule (Zhang et al., 2021a). Optionally, possibly incorrect labels can be flipped to the current model predictions (Zhang et al., 2021a).

We can find that such "hard" sample selection methods have two options for a training example: it is either completely reliable or not reliable at all. However, since mislabeled instances are more likely to be near the decision boundary, such hard decision rules may result in unstable training. To cope with this issue, they often requires the confidence regularization, warm-up training, or a manually designed threshold decay schedule, which introduces more hyperparameters such as the regularization weight, the number of warm-up epochs, and the threshold schedule.

In contrast, in our method, the confidence $C$ – the "soft" threshold — is continuous in $[0, 1]$ and is updated smoothly using the current prediction and gradient information so the training may be more stable. We do not need to introduce other hyperparameters. Although we do need to specify an optimizer for it, we found that a simple SGD optimizer works well enough in experiments.

Another benefit of using continuous weights $[0, 1]$ instead of discrete selections $\{0, 1\}$ is that we can naturally derive a total order of training examples, which can be used for other tasks.

**Gradient-based optimization vs. optimal solution.**   Next, let us focus on the soft sample selection. Note that the confidence $C$ might have an optimal solution and can be obtained directly. However, it is only optimal given the current predicted probability $p$. If the model is still under training and the estimation of $p$ is not good enough yet, it is not always beneficial to give a definite prediction of $C$, which may cancel out the advantage of the soft sample selection.

More specifically, to give a prediction of $C$, we have the following choices. Their characteristics are also listed below.

1. using a neural network
   - generalizes to unseen instances
   - depends on the trajectory of training, gradient-based optimization (heavy)
   - time: smooth, space: smooth
2. using instance embedding
   - does not generalize to unseen instances
   - depends on the trajectory of training, gradient-based optimization (light)
   - time: smooth, space: non-smooth
3. using a predefined rule based on the probability
   - generalizes to unseen instances (via the classifier)
   - does not depend on the trajectory of training, no optimization
   - time: smooth (soft)/non-smooth (hard), space: non-smooth

Here, "smoothness" means if the values change gradually during training (time) and if adjacent instances tend to have similar values (space). We can use gradient-based optimization for instance embedding so it is smooth in time (for stable training) and non-smooth in space (for instance-dependent noise). Mathematically, the proposed method takes accounts of not only gradient direction but also gradient magnitude. Intuitively, recording previous confidences $C$ allows the model to have "memory" so $C$ is not solely decided by the current prediction. For the above reasons, we prefer soft sample selection and gradient-based optimization.

# E EXPERIMENTS

## E.1 IMAGE CLASSIFICATION

**Data.** We used the MNIST,[1] Fashion-MNIST,[2] Kuzushiji-MNIST,[3] SVHN,[4] CIFAR-10, and CIFAR-100[5] datasets. The MNIST, Fashion-MNIST, Kuzushiji-MNIST datasets contain $28 \times 28$ grayscale images in 10 classes. The size of the training set is 60000 and the size of the test set is 10000. The SVHN dataset contains $32 \times 32$ colour images in 10 classes. The size of the training set is 73257 and the size of the test set is 26032. The CIFAR-10 and CIFAR-100 datasets contain $32 \times 32$ colour images in 10 classes and in 100 classes, respectively. The size of the training set is 50000 and the size of the test set is 10000. We used 20% of the training sets for validation. We added synthetic label noise into the training and validation sets. We used a method similar to Zhang et al. (2021a) (hybrid noise with 20% Type-I + 40% Uniform) to generate synthetic instance-dependent noise. Concretely, we first approximated the clean class-posterior using the original clean datasets, and corrupted the label from the most confident prediction to the second most confident one. The overall noise rate was around 52%, depending on the datasets. The test sets were not modified.

**Models.** For MNIST, Fashion-MNIST, and Kuzushiji-MNIST, we used a sequential convolutional neural network with the following structure: `Conv2d`(channel=32) $\times2$, `Conv2d`(channel=64) $\times2$, `MaxPool2d`(size=2), `Linear`(dim=128), `Dropout`(p=0.5), `Linear`(dim=10). The kernel size of convolutional layers is 3, and rectified linear unit (ReLU) is applied after the convolutional layers and linear layers except the last one. For SVHN, CIFAR-10 and CIFAR-100, we used a ResNet-18 model (He et al., 2016). To ensure that $C \in [0, 1]$, we simply apply the sigmoid function that maps $\mathbb{R}$ to $[0, 1]$ to the embedding.

**Optimization.** For MNIST, Fashion-MNIST, and Kuzushiji-MNIST, we used an Adam optimizer (Kingma & Ba, 2015) with batch size of 512 and learning rate of $1 \times 10^{-3}$. The model was trained for 2000 iterations (17.07 epochs) and the learning rate decayed exponentially to $1 \times 10^{-4}$. For CIFAR-10 and CIFAR-100, we used a stochastic gradient descent (SGD) optimizer with batch size of 512, momentum of 0.9, and weight decay of $1 \times 10^{-4}$. The learning rate increased from 0 to 0.1 linearly for 400 iterations and decreased to 0 linearly for 3600 iterations (4000 iterations/40.96 epochs in total). For SVHN, the setting was the same as CIFAR-10 except the model was trained for 1000 iterations.

**Results.** The ridgeline plots of the confidence $C$ during training are given in Fig. 10. The densities seem bimodal in the early stage for MNIST, FMNIST, and KMNIST because we did not log the result at the end of each epoch but based on the number of batch iterations. Therefore, the confidence of some instances may have been updated more times than it of the others. In the final stage, the confidence also converges and instances with flipped/original labels are almost separated by the learned confidence. Low-confidence training examples are given in Figs. 11 and 13, which are partially presented in Fig. 5. The class-wise spectra are given in Fig. 12, which are partially presented in Fig. 6.

## E.2 TEXT CLASSIFICATION

We implemented the BERT-base model (Devlin et al., 2019) using PyTorch (Paszke et al., 2019) and HuggingFace's `transformers` (Wolf et al., 2020) libraries. We used a pretrained model[6] and an AdamW optimizer (Loshchilov & Hutter, 2017). The batch size was 32 and the weight decay

---

[1] MNIST (LeCun et al., 1998) http://yann.lecun.com/exdb/mnist/

[2] Fashion-MNIST (Xiao et al., 2017) https://github.com/zalandoresearch/fashion-mnist (MIT license)

[3] Kuzushiji-MNIST (Clanuwat et al., 2018) http://codh.rois.ac.jp/kmnist/ (CC BY-SA 4.0 license)

[4] SVHN (Netzer et al., 2011) http://ufldl.stanford.edu/housenumbers

[5] CIFAR-10, CIFAR-100 (Krizhevsky, 2009) https://www.cs.toronto.edu/~kriz/cifar.html

[6] `bert-base-cased`: https://huggingface.co/bert-base-cased

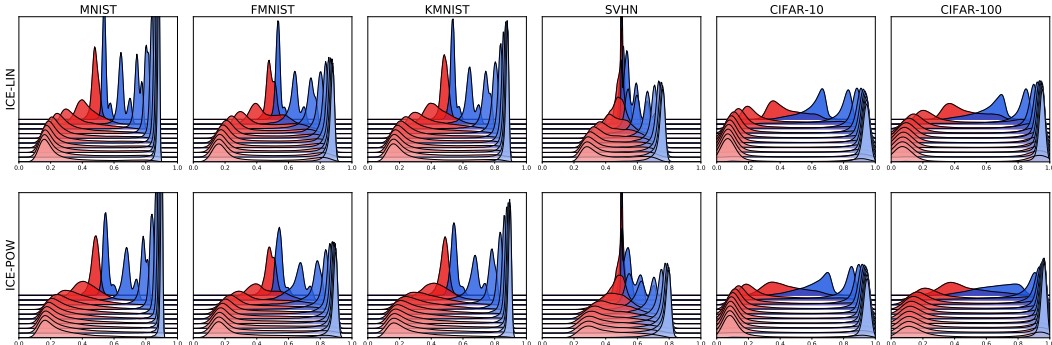

Figure 10: **Ridgeline plots** of the confidence $C$ during training. The density is estimated via Gaussian kernel density estimation (KDE). The red/blue curves represent the confidence of instances with flipped/original labels, respectively.

was 0.01, otherwise we used the default hyperparameters. The model was trained on 4 NVIDIA Tesla P100 GPUs in parallel with the mixed precision training option (`fp16`) enabled. For the CoLA, MRPC, RTE, and WNLI datasets, the model was trained for 5 epochs and otherwise 3 epochs. Low-confidence training examples are given in Tables 6 to 13.

### E.3 MORE EXPERIMENTAL RESULTS

We also conducted the experiment on a common setting where the labels are randomly flipped, i.e., instance-independently. The noise rate was 50%. The test accuracy is listed in Table 4.

Table 4: **Accuracy** (%) on the MNIST, FMNIST, KMNIST, SVHN, CIFAR-10, and CIFAR-100 datasets where 50% of labels are randomly flipped. "Mean (standard deviation)" for 10 trials are reported. Outperforming methods are highlighted in boldface using one-tailed t-tests with a significance level of 0.05.

|  | MNIST | FMNIST | KMNIST | SVHN | CIFAR-10 | CIFAR-100 |
|---|---|---|---|---|---|---|
| CCE | 94.91(0.43) | 85.05(0.52) | 80.40(1.25) | 71.50(1.68) | 68.34(0.82) | 47.09(0.65) |
| Bootstrapping | 97.30(0.28) | 87.24(0.36) | 84.21(1.01) | 76.62(0.97) | 75.97(0.45) | 49.56(0.42) |
| Adaptation | 96.27(0.41) | 86.20(0.87) | 81.38(2.07) | 68.58(6.45) | 63.95(5.94) | 31.70(1.28) |
| Forward | 95.09(0.56) | 85.51(0.45) | 80.76(1.29) | 74.43(6.42) | 68.28(0.62) | 47.92(0.31) |
| Dual-T | 97.68(0.25) | 88.07(0.35) | 86.98(0.80) | 72.36(0.88) | 78.53(0.32) | 50.80(0.40) |
| DAC | 96.60(0.47) | 86.87(0.48) | 82.77(0.74) | **80.97(4.83)** | 71.55(0.34) | 47.01(0.44) |
| GCE | 98.31(0.13) | 88.76(0.26) | 88.39(0.60) | 75.03(0.98) | 80.38(0.67) | **55.64(0.40)** |
| ICE-LIN | **98.64(0.15)** | **89.41(0.18)** | **89.61(0.41)** | 77.51(0.75) | **82.08(0.39)** | **55.30(0.47)** |
| ICE-POW | **98.60(0.09)** | **89.29(0.20)** | 89.21(0.53) | **79.91(0.96)** | **82.14(0.44)** | 54.31(0.48) |

Further, we tested the proposed method on the Clothing1M dataset (Xiao et al., 2015), which is a real-world noisy label dataset. We followed the convention and trained a ResNet-50 model (He et al., 2016) only on the 1M noisy training set for 10 epochs. On this dataset, ICE-POW achieved the accuracy of 72.67%, which is comparable or superior to some existing works. For reference, the reported performances of other methods including CCE, Forward, T-Revision (Xia et al., 2019), dual-T (Yao et al., 2020), PTD (Xia et al., 2020), DMI (Xu et al., 2019), CORES[2] (Cheng et al., 2021), ILFC (Berthon et al., 2021), and PLC (Zhang et al., 2021a), are cited in Table 5.

Table 5: Test accuracy on Clothing1M.

| CCE | Forward | T-Revision | Dual-T | PTD | DMI | CORES[2] | ILFC | PLC |
|---|---|---|---|---|---|---|---|---|
| 68.94 | 70.83 | 70.97 | 71.49 | 71.67 | 72.46 | 73.24 | 73.35 | 74.02 |

Note that some methods are superior to ours in terms of accuracy on Clothing1M. This may be due to different evaluation settings and training techniques. Further investment is left for future work.

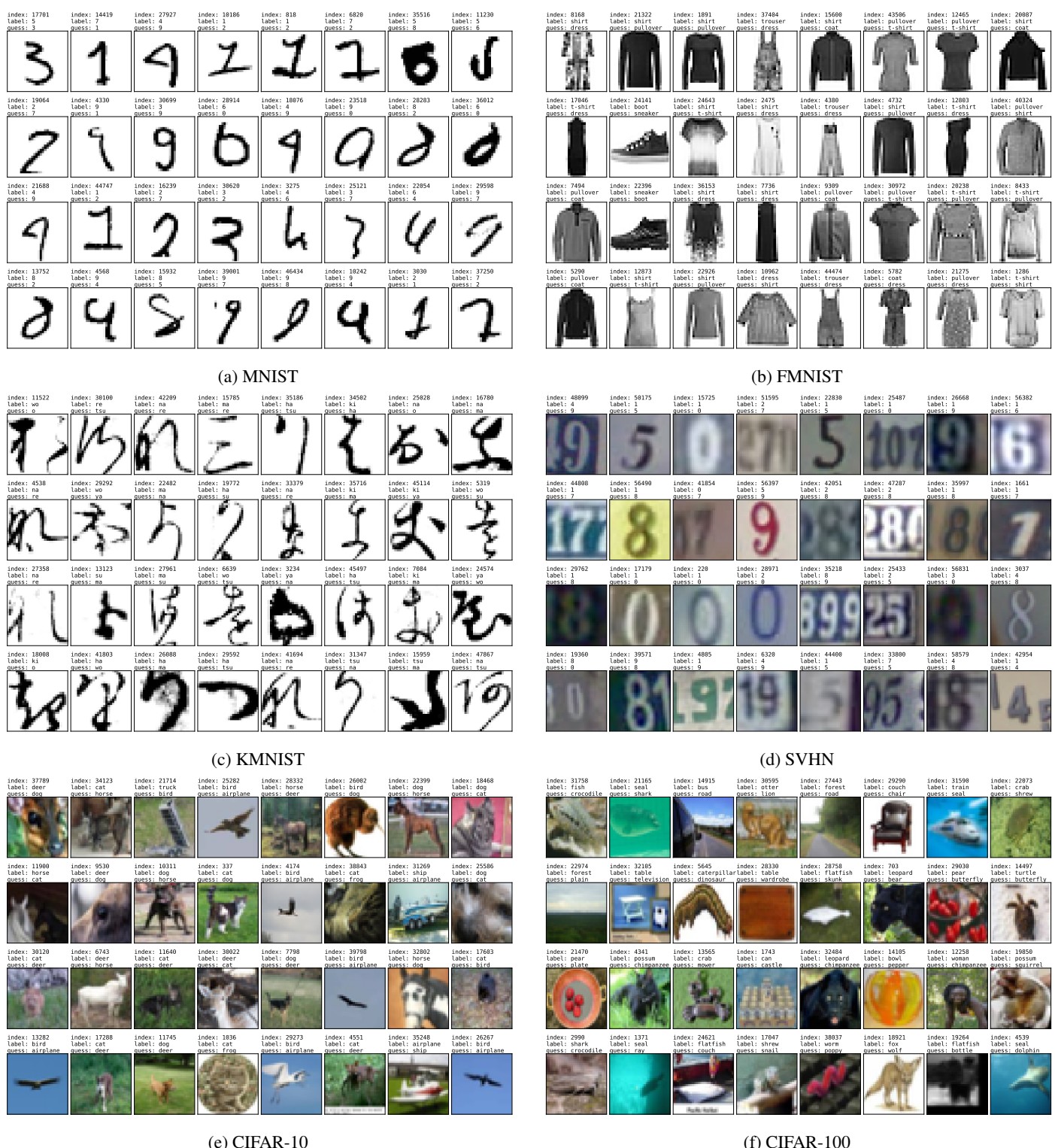

Figure 11: The 32 most **low-confidence training examples** in the MNIST, FMNIST, KMNIST, SVHN, CIFAR-10 and CIFAR-100 datasets, ordered left-right, top-down by increasing confidence.

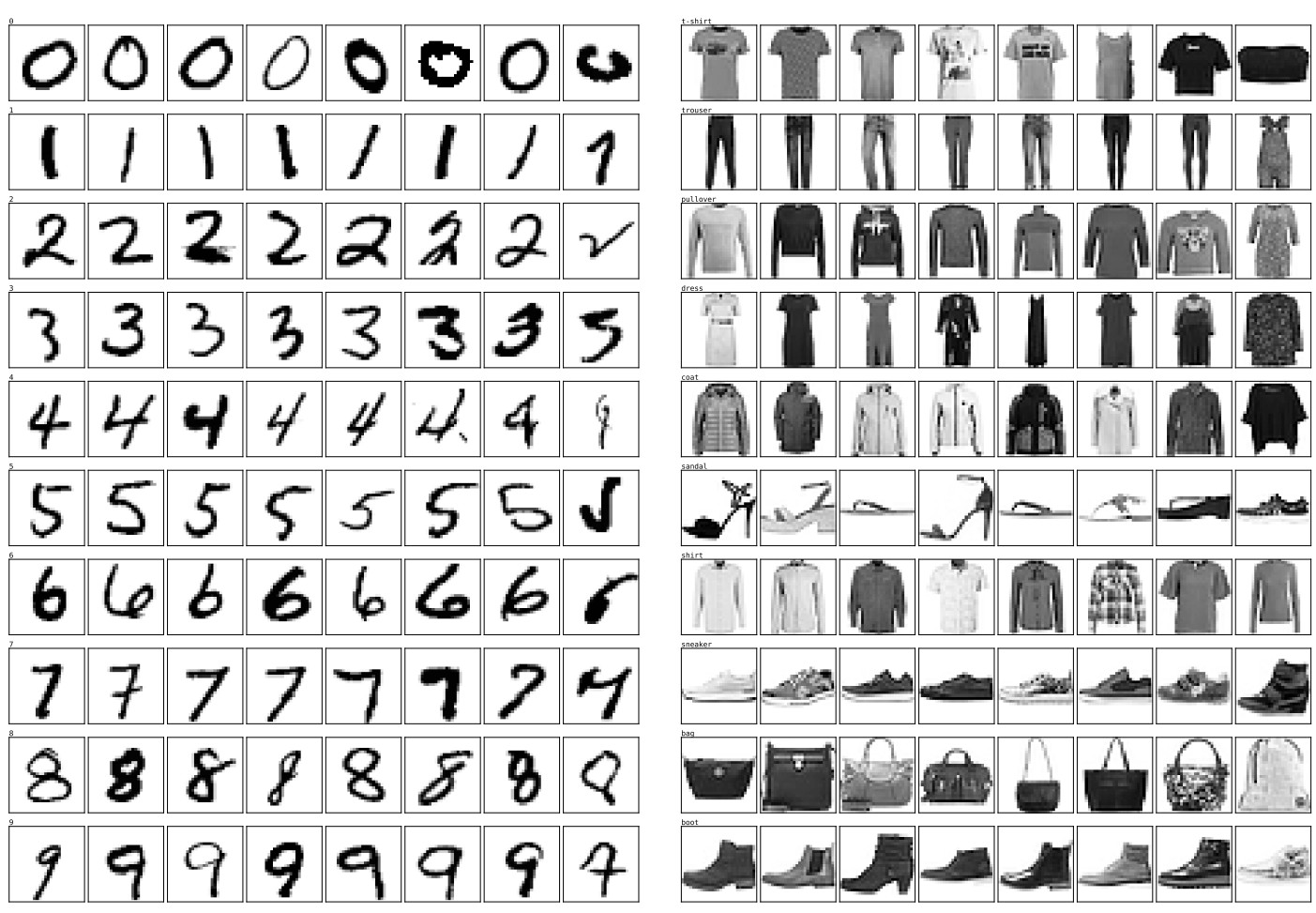

(a) MNIST

(b) FMNIST

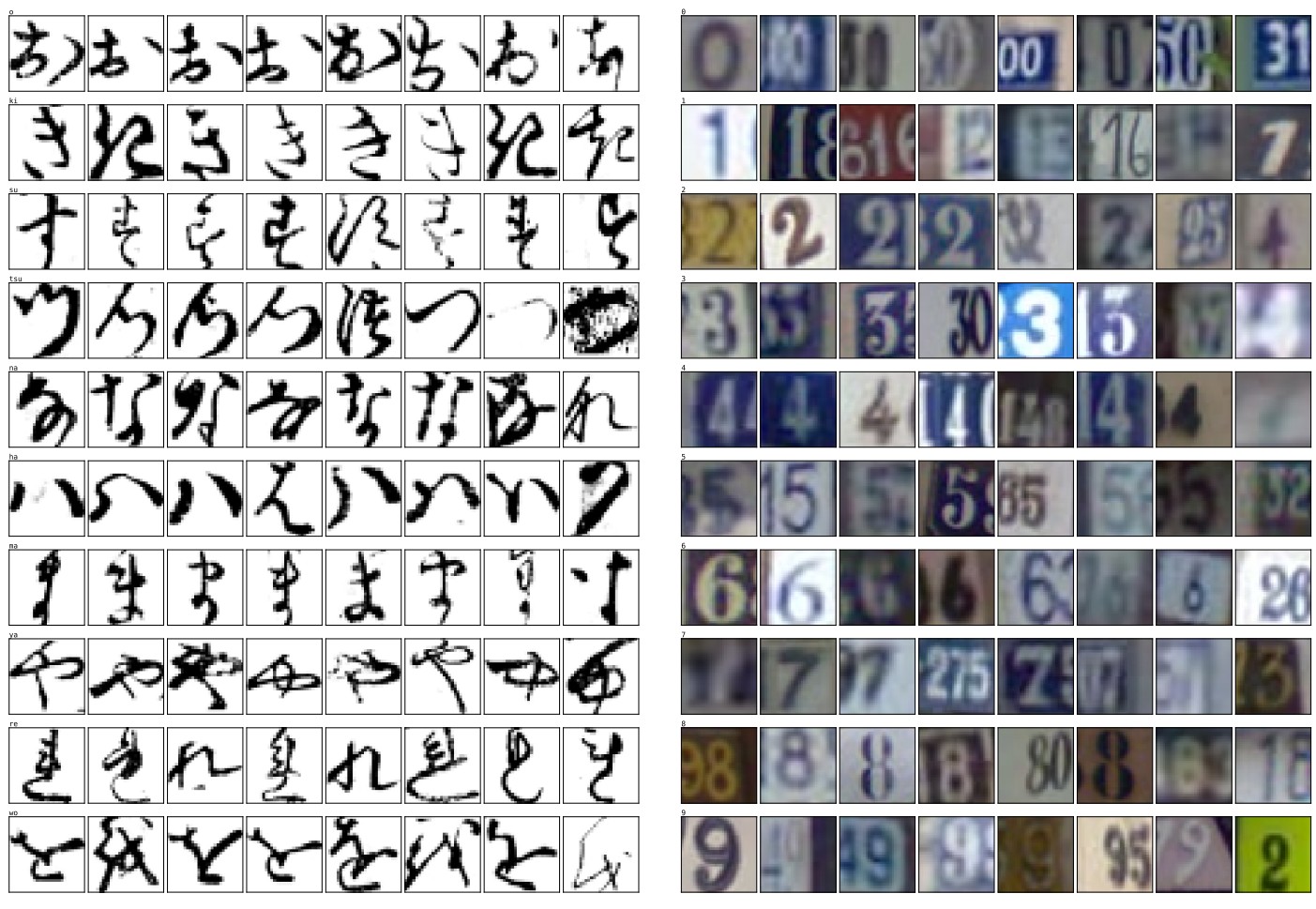

(c) KMNIST                              (d) SVHN

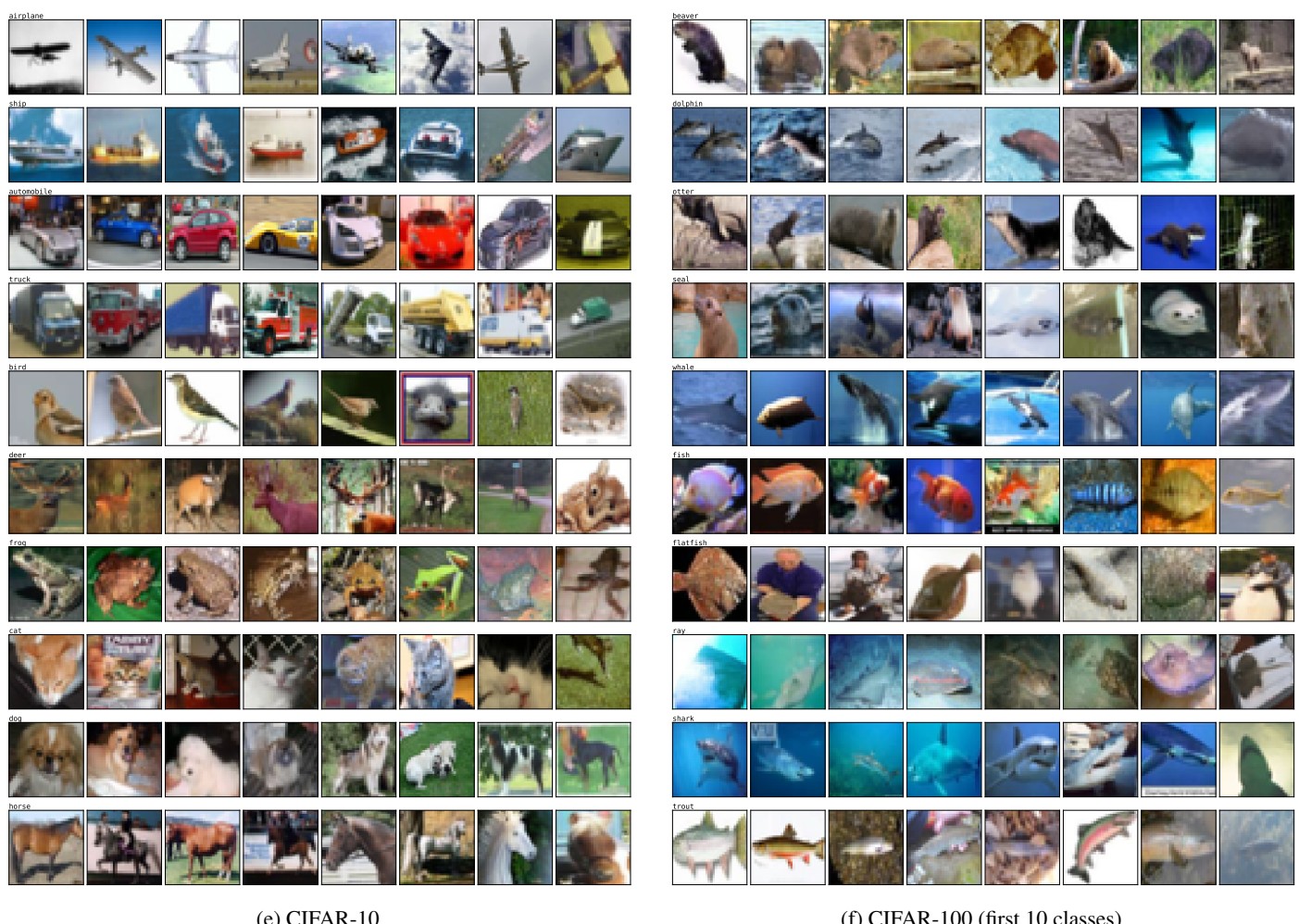

(e) CIFAR-10                        (f) CIFAR-100 (first 10 classes)

Figure 12: The **class-wise spectra** from high-confidence (left) to low-confidence (right) training examples in the MNIST, FMNIST, KMNIST, SVHN, CIFAR-10 and CIFAR-100 datasets.

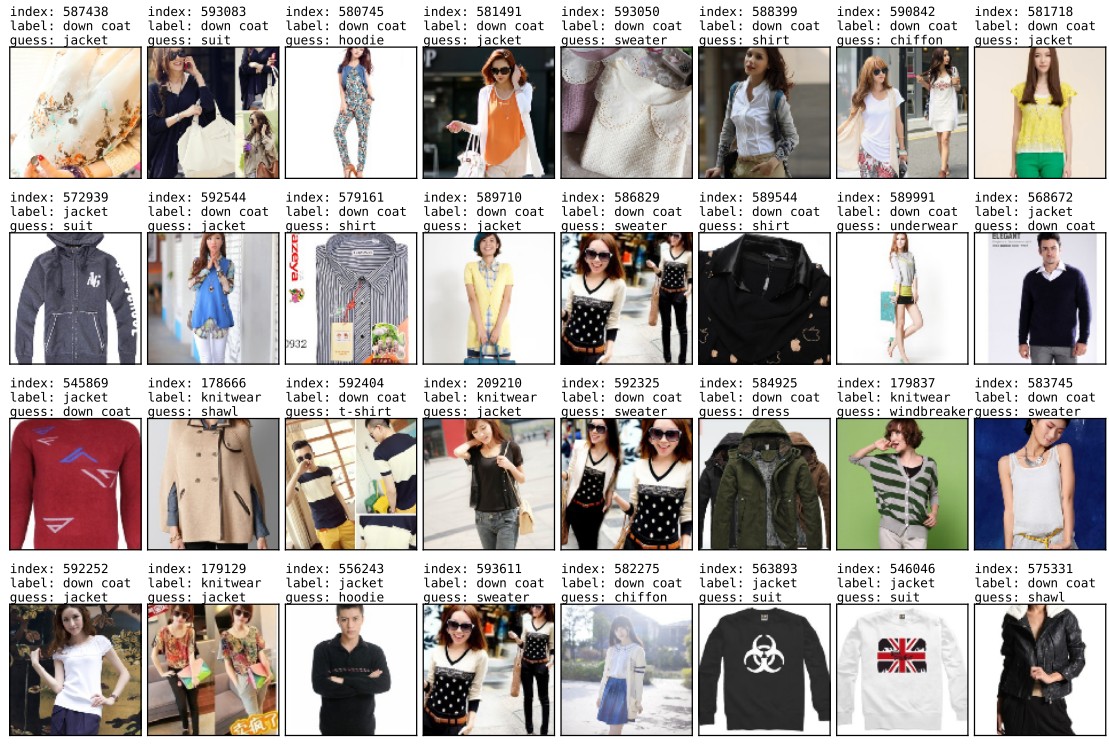

Figure 13: The 32 most **low-confidence training examples** in the Clothing1M dataset, ordered left-right, top-down by increasing confidence.

Table 6: The Corpus of Linguistic Acceptability (CoLA)

| Index | Text |
|---|---|
| 390 | label:acceptable guess:unacceptable
sentence:He I often sees Mary. |
| 5766 | label:acceptable guess:unacceptable
sentence:Heidi believes any description of herself. |
| 2801 | label:unacceptable guess:acceptable
sentence:Paula hit the sticks. |
| 1522 | label:unacceptable guess:acceptable
sentence:That the sun is out was obvious. |
| 8332 | label:acceptable guess:unacceptable
sentence:I wanted Jimmy for to come with me. |
| 300 | label:acceptable guess:unacceptable
sentence:They failed to tell me which problem the sooner I solve, the quicker the folks up at corporate headquarters. |
| 7813 | label:acceptable guess:unacceptable
sentence:I went to the shop for to get bread. |
| 5904 | label:acceptable guess:unacceptable
sentence:It hailed. |
| 4159 | label:unacceptable guess:acceptable
sentence:Fifteen years represent a long period of his life. |
| 2479 | label:unacceptable guess:acceptable
sentence:Kelly buttered the bread with butter. |
| 3846 | label:acceptable guess:acceptable
sentence:They parted the best of friends. |
| 7371 | label:unacceptable guess:acceptable
sentence:The hiker will reach the top of the mountain for an hour. |
| 430 | label:unacceptable guess:acceptable
sentence:It's probable in general that he understands what's going on. |
| 6795 | label:unacceptable guess:acceptable
sentence:Henry wanted to possibly marry Fanny. |
| 1115 | label:acceptable guess:acceptable
sentence:He attributed to a short circuit the fire which. |
| 4155 | label:unacceptable guess:unacceptable
sentence:Two drops sanitize anything in your house. |
| 1367 | label:acceptable guess:acceptable
sentence:We elected president the boy's guardian's employer. |
| 7756 | label:acceptable guess:unacceptable
sentence:That monkey is ate the banana |
| 4445 | label:unacceptable guess:acceptable
sentence:George has went to America. |
| 4015 | label:acceptable guess:unacceptable
sentence:He seems intelligent to study medicine. |

Table 7: The Stanford Sentiment Treebank (SST2)

| Index | Text |
|---|---|
| 50155 | label:positive guess:positive
sentence:a thirteen-year-old 's book report |
| 58416 | label:negative guess:negative
sentence:blues |
| 59724 | label:negative guess:positive
sentence:' synthetic ' is the best description of this well-meaning ,
beautifully produced film that sacrifices its promise for a high-powered star
pedigree . |
| 24696 | label:positive guess:negative
sentence:lamer instincts |
| 34494 | label:positive guess:positive
sentence:had released the outtakes theatrically and used the film as a bonus
feature on the dvd |
| 54555 | label:negative guess:negative
sentence:pretentious , fascinating , ludicrous , provocative and vainglorious |
| 29155 | label:positive guess:negative
sentence:he can be forgiven for frequently pandering to fans of the gross-out
comedy |
| 44610 | label:negative guess:positive
sentence:below |
| 66148 | label:negative guess:positive
sentence:'s cliche to call the film ' refreshing |
| 11869 | label:positive guess:negative
sentence:go unnoticed and underappreciated |
| 55848 | label:negative guess:positive
sentence:the film is an earnest try at beachcombing verismo , but it would be
even more indistinct than it is were it not for the striking , quietly
vulnerable personality of ms. ambrose . |
| 57359 | label:negative guess:negative
sentence:( ferrera ) |
| 42232 | label:positive guess:negative
sentence:forgive any shoddy product as long as there 's a little girl-on-girl
action |
| 15783 | label:positive guess:negative
sentence:have finally aged past his prime ... |
| 57186 | label:negative guess:positive
sentence:hollywood war-movie stuff |
| 52071 | label:positive guess:negative
sentence:the gags |
| 1896 | label:positive guess:negative
sentence:missing from the girls ' big-screen blowout |
| 64779 | label:positive guess:negative
sentence:growing strain |
| 3940 | label:positive guess:negative
sentence:you to bite your tongue to keep from laughing at the ridiculous dialog
 or the oh-so convenient plot twists |

Table 8: Microsoft Research Paraphrase Corpus (MRPC)

| Index | Text |
|---|---|
| 799 | label:equivalent guess:not equivalent
sentence1:We need a certifiable pay as you go budget by mid-July or schools wont open in September , Strayhorn said .
sentence2:Texas lawmakers must close a $ 185.9 million budget gap by the middle of July or the schools wont open in September , Comptroller Carole Keeton Strayhorn said Thursday . |
| 469 | label:not equivalent guess:equivalent
sentence1:It 's also a strategic win for Overture , given that Knight Ridder had the option of signing on Google 's services .
sentence2:It 's also a strategic win for Overture , given that Knight Ridder had been using Google 's advertising services . |
| 1037 | label:equivalent guess:not equivalent
sentence1:The broader Standard & Poor 's 500 Index < .SPX > edged down 9 points , or 0.98 percent , to 921 .
sentence2:The Standard & Poor 's 500 Index shed 5.20 , or 0.6 percent , to 924.42 as of 9 : 33 a.m. in New York . |
| 1178 | label:equivalent guess:not equivalent
sentence1:Sens. John Kerry and Bob Graham declined invitations to speak .
sentence2:The no-shows were Sens. John Kerry of Massachusetts and Bob Graham of Florida . |
| 1753 | label:equivalent guess:not equivalent
sentence1:The Dow Jones industrial average closed down 18.06 , or 0.2 per cent , at 9266.51 .
sentence2:The blue-chip Dow Jones industrial average < .DJI > slipped 44.32 points , or 0.48 percent , to 9,240.25 . |

Table 9: Quora Question Pairs (QQP)

| Index | Text |
|---|---|
| 216515 | label:duplicate guess:duplicate
question1:Why does Quora censor opinions and answers?
question2:Does Quora censor questions and answers, and should they? |
| 343656 | label:not duplicate guess:duplicate
question1:Could India's surgical strike in POK be an elaborate hoax or play?
question2:Did India really conduct a surgical strike on Pakistan? |
| 266594 | label:not duplicate guess:not duplicate
question1:Why is financial literacy generally not taught in American high schools?
question2:Why isn't financial literacy taught in today's public schools? |
| 251996 | label:not duplicate guess:not duplicate
question1:What is life like in communist countries?
question2:What would life in a legitimate Communist country be like? |
| 7963 | label:not duplicate guess:duplicate
question1:What is the best option for Indian politics and politicians?
question2:What are the options of Indian politics and politicians? |

## Table 10: MultiNLI (MNLI)

| Index | Text |
|---|---|
| 218290 | label:entailment guess:neutral
premise:The ruins of the huge abbey of Jumiyges are perhaps the most the white-granite shells of two churches, the Romanesque Notre-Dame and the smaller Gothic Saint-Pierre.
hypothesis:Notre-Dame is a larger church than Gothic Saint-Pierre. |
| 39431 | label:neutral guess:entailment
premise:Unless you feel really safe in French metropolitan traffic, keep your cycling ' you can rent a bike at many railway stations ' for the villages and country roads.
hypothesis:You should not cycle in the French metropolitan area. |
| 27574 | label:contradiction guess:neutral
premise:I don't think so.
hypothesis:I have no real idea. |
| 258544 | label:neutral guess:neutral
premise:A set of stone doors in the wall slid to the side to reveal a screen on which various torture scenes began to appear.
hypothesis:The doors hid a television screen. |
| 320518 | label:contradiction guess:neutral
premise:None seems comfortable with the notion of removing Clinton for sex-related misdeeds.
hypothesis:People don't want Clinton touching sex related ordeals |

## Table 11: Question NLI (QNLI)

| Index | Text |
|---|---|
| 1659 | label:not entailment guess:not entailment
question:What caused Latin America's right-wing authorities to support coup o' etats?
sentence:This was further fueled by Cuban and United States intervention which led to a political polarization. |
| 5876 | label:not entailment guess:not entailment
question:What antenna type is a portion of the half wave dipole?
sentence:The monopole antenna is essentially one half of the half-wave dipole, a single 1/4-wavelength element with the other side connected to ground or an equivalent ground plane (or counterpoise). |
| 5829 | label:entailment guess:entailment
question:How are Toxicara canis infections spread?
sentence:Toxocara canis (dog roundworm)eggs in dog feces can cause toxocariasis. |
| 77419 | label:entailment guess:entailment
question:Why did Madrid cede the territory to the US
sentence:Florida had become a burden to Spain, which could not afford to send settlers or garrisons. |
| 9576 | label:not entailment guess:not entailment
question:What has no distinction between the categories of voiced, voiceless, aspirated and unaspirated?
sentence:Some of the Dravidian languages, such as Telugu, Tamil, Malayalam, and Kannada, have a distinction between voiced and voiceless, aspirated and unaspirated only in loanwords from Indo-Aryan languages. |

Table 12: Recognizing Textual Entailment (RTE)

| Index | Text |
|-------|------|
| 2429 | label:not entailment guess:entailment
sentence1:Bogota, 4 May 88 – The dissemination of a document questioning Colombia's oil policy, is reportedly the aim of the publicity stunt carried out by the pro-Castro Army Of National Liberation, which kidnapped several honorary consuls, newsmen, and political leaders.
sentence2:Several honorary consuls were kidnapped on 4 May 88. |
| 2463 | label:not entailment guess:entailment
sentence1:The official religion is Theravada Buddhism, which is also practiced in neighboring Laos, Thailand, Burma and Sri Lanka.
sentence2:The official religion of Thailand is Theravada Buddhism. |
| 1361 | label:entailment guess:not entailment
sentence1:The Catering JLC formulates pay and conditions proposals of workers in the industry which, if approved by the Labour Court, legally binds employers to pay certain wage rates and provide conditions of employment. However,the QSFA now contends that the JLC has no right to make such a legally binding provision, as Section 15 of the Constitution states that the sole and exclusive power to make laws is vested in the Oireachtas, and no other authority has power to make laws for the State. It also argued that the existence of the minimum wage and 25 other pieces of legislation protecting employees' rights means that there is no need for JLCs. The chairman of the QSFA, John Grace, warned that the situation would lead to job losses and closures.
sentence2:John Grace works for QSFA. |

Table 13: Winograd NLI (WNLI)

| Index | Text |
|-------|------|
| 266 | label:entailment guess:not entailment
sentence1:Susan knew that Ann's son had been in a car accident, so she told her about it.
sentence2:Susan told her about it. |
| 478 | label:entailment guess:not entailment
sentence1:Joe paid the detective after he delivered the final report on the case.
sentence2:The detective delivered the final report on the case. |
| 294 | label:entailment guess:not entailment
sentence1:Dan had to stop Bill from toying with the injured bird. He is very compassionate.
sentence2:Dan is very compassionate. |
| 586 | label:entailment guess:not entailment
sentence1:Dan took the rear seat while Bill claimed the front because his "Dibs!" was slow.
sentence2:Dan took the rear seat while Bill claimed the front because Dan's "Dibs!" was slow. |
| 243 | label:entailment guess:not entailment
sentence1:Mark was close to Mr. Singer's heels. He heard him calling for the captain, promising him, in the jargon everyone talked that night, that not one thing should be damaged on the ship except only the ammunition, but the captain and all his crew had best stay in the cabin until the work was over.
sentence2:He heard Mr. Singer calling for the captain. |

