# OpenReview forum: "Approximating Instance-Dependent Noise via Instance-Confidence Embedding"
_ICLR.cc/2022/Conference — ICLR 2022 Submitted_

### Official Review · Reviewer_ARcm · 2021-10-24

**Correctness:** 3
**Technical Novelty And Significance:** 2
**Empirical Novelty And Significance:** 2
**Recommendation:** 5
**Confidence:** 4

**Main Review:**

Strength:
1. This paper is well written and easy to follow. The motivation is strong and interesting. It is of great importance to approximate IDN with a simple method since IDN is generally hard to model. This paper proposes a simple way to approximate IDN by using single-scalar confident parameter.

Concerns:
1. I am quite confused by the part of embedding. The embedding is built on the sample index instead of the sample (feature) itself. I think it does not follow the definition of IDN. $C$ in Equation (8) (9) should be dependent on the feature rather than the feature index.

2. Many methods have been proposed in the literature to deal with IDN (A1, A2, A3, A4, A5). However, in Table 1, the authors do not compare any of them. Besides, the performance gain of ICE is minor compared to GCE on CIFAR10 and CIFAR100.

3. ICE is similar to label smoothing in the formulation. Authors are encouraged to perform a comparison in the experiments.

4. Why only perform 40 epochs on CIFAR10 on CIFAR100. It does not follow the convention. Does the network begin to overfit label noise as training proceeds?

A1: Learning with bounded instance- and label-dependent label noise. ICML 2020

A2: Part-dependent label noise: Towards instance-dependent label noise. NeurlPS 2020

A3: Confidence scores make instance-dependent label-noise learning possible. ICML2021

A4: Learning with feature-dependent label noise: A progressive approach. ICLR2021

A5: Learning with instance-dependent label noise: A sample sieve approach. ICLR2021

**Summary Of The Paper:**

This paper proposes ICE (Instance-confident embedding) method for learning with IDN (instance-dependent label noise). ICE approximates IDN by using a single-scalar confident parameter for each sample based on the assumption that $P(Y|x)$ is close to a one-hot vector. ICE is incorporated into the loss function for efficient learning process under label noise. Authors conduct experiments on both image and text classification to verify the effectiveness  of ICE and perform gradient analysis for further understanding the method.


**Summary Of The Review:**

This paper has a strong motivation. However, it seems that the proposed approach (ICE) does not strongly connect to IDN and the performance gain is minor compared to existing approaches. Authors should provide more explanations of why embedding on the index can approximate IDN and perform more comparisons.

---

> ### Author Response · Authors · 2021-11-18
> **Response to Reviewer ARcm**
>
> Thank you for reviewing our paper and pointing out a confusing part of instance embedding.
>
> We would like to address your concerns as follows.
>
> ### Instance embedding
>
> We agree that it might be slightly counterintuitive to only use the index but not the feature itself.
> However, to be more mathematically rigorous, we can see that it is still a valid approach, which is shown below.
>
> Our goal is to find a mapping $g: \mathcal{X} \to [0, 1]$.
> We are given a finite i.i.d. sample $\bar{\mathcal{X}} = \\{x_i\\}_{i=1}^N \subset \mathcal{X}$ of size $N$ drawn from some distribution $x_i \sim p(X)$.
>
> If we want to find a mapping that generalizes to points other than those in $\bar{\mathcal{X}}$, we need some inductive biases (model, risk, etc.) to find a mapping that is optimal in some sense.
> Usually, we can use a family of mappings parameterized by $\theta \in \Theta$, such as neural networks, and find the optimal parameter $\hat{\theta}$.
> In this way, when an unseen instance comes, we can still use the optimal parameter to predict the target.
>
> However, in our case, we found the generalization ability of $g$ is not needed.
> By **giving up the generalization ability** of $g$, we only need to find a **restriction** $g|_{\bar{\mathcal{X}}}: \bar{\mathcal{X}} \to [0, 1]$ of $g$ to $\bar{\mathcal{X}}$.
> Then, its domain becomes a finite set $\bar{\mathcal{X}}$.
>
> To find a mapping whose domain is a finite set, we can use the following parameterized mappings:
> \begin{equation}
> g_{\mathrm{embed}}(x; \theta) := \sum_{i=1}^N 1[x = x_i] \cdot s(\theta_i)
> \end{equation}
> where $1[\cdot]$ denotes the **indicator function/Iverson bracket**, and $s: \mathbb{R} \to [0, 1]$ maps an unconstrained parameter $\theta_i \in \mathbb{R}$ to the codomain $[0, 1]$ (In our case $s := \theta_i \mapsto C_i$ and we use a sigmoid function).
> Here, $g_{\mathrm{embed}}$ is parameterized by $\theta \in \mathbb{R}^N$, i.e., $N$ scalar values.
>
> We can see that it is still a function of $x$, although actually only its index in the sample is needed.
> **We want $g$ to be dependent on the feature only if we want it to generalize to other points**.
>
> In this formulation, $g_{\mathrm{embed}}$ has maximal expressiveness (all possible mappings, but only on $\bar{\mathcal{X}}$), but zero generalization ability.
> We only need $f$ to generalize to other instances, because by our design, the decision based on $h(f(x), g(x))$ during training is the same as the decision based on only $f(x)$ during test when we do not need (and do not have) the confidence parameters.
> We think that showing "**we can replace mappings that do not need to generalize to unseen instances with trainable parameters for each training instance**" is one of the contributions of this work.
>
> We hope this clarifies your confusion.
>
> ### Performance
>
> Thank you for your suggestion!
>
> To improve the credibility, we have added [A4] PLC (Zhang et al. (2021a)) to Table 1 in the revised paper.
> We found that PLC performs well, especially in SVHN (PLC $86.23\\%$ vs. ICE-POW $81.82\\%$).
> We hypothesize that it is due to images with multiple digits in SVHN (see also Figures 11 and 12 in Appendix E).
> Meanwhile, ICE still performs comparably to PLC on other datasets.
>
> We have not compared with [A1] because it is mainly for binary classification (although extensions may be possible).
> We have not compared with [A2] (Part-dependent Label Noise) and [A3] (Confidence Score) because they either require domain-specific knowledge or extra supervision, which was discussed in the introduction.
> To avoid such limits is one of our motivations to do this research.
>
> As for the performance, we admit that some more complex methods may perform better on some datasets, which we have cited and discussed in Appendix E.
> However, we found that such a simple method is usually comparable to the state-of-the-art approaches, which might be useful to the community.
>
> Further, we would like to argue that "performance gain" in terms of test accuracy is not the only criterion, and we would highlight the extra benefits of ICE that learned confidence parameters may be informative and useful for some tasks, such as ranking the reliability of training examples and detecting mislabeled instances (Figure 5, Table 3, and more in Appendix E).
> We also look forward to its potential applications in data cleansing, learning with rejection, or active learning.
> This is why we would like to share this idea with the community.

---

> > ### Author Response · Authors · 2021-11-18
> > **Response to Reviewer ARcm**
> >
> > ### Others
> >
> > ICE-LIN and label smoothing both use linear interpolation, as well as the bootstrapping loss that we experimentally evaluated.
> > However, the formulations are actually different: label smoothing/bootstrapping interpolates the label, while ICE-LIN interpolates the prediction, which was discussed in Section 4 related work and further visualized in Appendix C linear interpolation.
> >
> > It is arguable that training with a lot of epochs is a convention.
> > The number of epochs usually relies on other hyper-parameters.
> > We think a proper practice is to train until convergence.
> > We have conducted experiments with more epochs and no evidence of overfitting was shown.

---

> > ### Author Response · Authors · 2021-11-20
> > **Word Embedding vs. Instance-Confidence Embedding**
> >
> > ### Embedding
> >
> > Additionally, let us compare word embedding used in NLP and instance-confidence embedding used here, which was partially discussed in Section 3.3.
> >
> > |                | Word Embedding                                                | Instance-Confidence Embedding                                                                 |
> > |----------------|---------------------------------------------------------------|-----------------------------------------------------------------------------------------------|
> > | Feature        | alphabet, length, ... not so informative about word's meaning | instance itself, not so informative about how confident the classifier should be                 |
> > | Supervision    | the vector embedding of a word is not given as supervision  | the confidence parameter of an instance is not given as supervision                            |
> > | Training       | the meaning of a word is determined by its context            | the confidence for an instance is determined by the prediction of adjacent instances          |
> > | Generalization | usually no need to generalize to unknown tokens               | usually no need to generalize to unseen instances because the decision does not require this information |
> >
> > We hope this helps to contextualize and justify the use of embedding.
> >
> > Thank you for bringing up this point, which helps us to deepen our understanding of this approach.

---

> > ### Comment · Reviewer_ARcm · 2021-11-30
> > **Further concerns**
> >
> > I thank the authors' reply that responds to my review. I think ICE is indeed an interesting method with a strong motivation. However, after reading the newly revised submission and other reviews, I still have some further concerns.
> >
> > - After the authors' explanation, I understand that $g_{embed}$ can be regarded as a function of feature $x$. However, this relation (function) to $x$ is relatively weak to me since $g_{embed}$ is meaningful only when the feature $x$ equals to one of (N) limited points in the feature space. Considering the high dimensional feature space of $x$, I have a little doubt that one-single scalar parameter can well capture the instance-level based information. Is there any justification that can be made to calculate approximation error, experimentally or theoretically?
> >
> > - The major concern comes from experiments. I think if the experiments are very solid and relatively good, we can neglect some issues including approximation error brought by one-single parameter, unreasonable assumption of argmax-preserving because such method is simple, novel with a strong motivation which is worth being heard in the community. However, I don't think current experiments are sufficient. First, only one IDN method (PLC) is compared, and such results were added at the time of rebuttal. I think it is very necessary to compare more IDN methods in the next submission.
> > Second, the performance of ICE is not promising even compared to GCE which is not a method specifically designed for IDN. Due to the above reasons, I am skeptical about the applicability of ICE.
> >
> >
> > Additional comments:
> >
> > - I am still not sure why only performs 40 epochs on CIFAR. I understand that proper practice is to train until convergence. However, from my experience, if the dataset is clean or with little label noise, 40 epochs are not enough to let DNN have the best performance on validation or test set. In the literature of learning with noisy labels, many methods will train DNN on CIFAR with longer epochs and observe the convergence and over-fitting phenomenon. I suggest the authors add this part of the experiment.
> >
> > - I know that ICE is not equal to label smoothing (LS). However, the formulation is similar (LS smooths the target while ICE smooths the prediction). That's why I suggest the authors perform a comparison because LS is also an effective method for noisy label problems.

---

### Official Review · Reviewer_4pht · 2021-11-02

**Correctness:** 3
**Technical Novelty And Significance:** 4
**Empirical Novelty And Significance:** 3
**Recommendation:** 5
**Confidence:** 4

**Main Review:**

$\textbf{Strengths}$
* **Novelty** This work provides a novel and interesting approach to using only a single-scalar confidence parameter to approximate the IDN transition matrix.
* **Ambiguous/Noisy-label detection** The 1-D instance embedding is potentially beneficial for detecting ambiguous or wrongly labeled instances.
* **Extensive empirical validation** The authors test the effectiveness of ICE in multiple image classification datasets and text classification tasks.

$\textbf{Weaknesses}$
* **A single-scalar V.S. $K\times K$ noise transition matrix** Admittedly, the $K\times K$ noise transition matrix is usually of a sparse pattern. Will a single-scalar miss crucial information? Or in other words, could authors explain why this approximation captures necessary/useful noisy-label information at the instance level?

* **The intuition of $g$** It would be better if the authors could provide me with more intuitions of $g$ in the role of confidence score.

* **IDN-based baselines**  Learning with instance-dependent noisy labels has become a popular topic in recent years. And several approaches have been proposed to mitigate the impact of IDN. However, the majority of baselines adopted in Table 1 are designed for the CDN model. To further validate the effectiveness of ICE, it would be better if the authors could compare with some more IDN-based methods, for example, [1]-[5].

* **Real-world noisy benchmarks** Evaluating the rationality of (synthetic) instance-dependent label noise is non-trivial. Real-world noisy labels are commonly assumed to follow the IDN pattern. It would be better if the authors could test the performances on real-world noisy benchmarks, especially some small-scale benchmarks such as Animal-10N [6], Controlled Noisy Web Labels [7], or CIFAR-10N, CIFAR-100N [8].


$\textbf{Reference}$

[1] Part-dependent Label Noise: Towards Instance-dependent Label Noise. (NeurIPS, 2020)

[2] Learning with Feature-Dependent Label Noise: A Progressive Approach. (ICLR, 2021)

[3] Learning with Instance-Dependent Label Noise: A Sample Sieve Approach. (ICLR, 2021)

[4] A Second-Order Approach to Learning With Instance-Dependent Label Noise. (CVPR, 2021)

[5] Confidence Scores Make Instance-dependent Label-noise Learning Possible. (ICML, 2021)

[6] https://dm.kaist.ac.kr/datasets/animal-10n/.

[7] https://ai.googleblog.com/2020/08/understanding-deep-learning-on.html.

[8] http://noisylabels.com/.

**Summary Of The Paper:**

This paper introduces ICE, an instance-confidence embedding approach to learn with a challenging noisy label setting: instance-dependent noise (IDN). Considering the sparse property of IDN, the authors propose to use instance embedding by equipping a trainable parameter to each data instance. Experiments results on various image datasets and text data show the effectiveness of the proposed method with the presence of label noise.

**Summary Of The Review:**

This is an overall novel and interesting paper. While I still have some concerns regarding the single-scale confidence parameter and the intuition of the instance-confidence embedding function $g$. Besides, I am not fully convinced of the effectiveness of ICE by referring to the baseline selections. I am willing to increase my score if some of my above concerns are well addressed.

---

> ### Author Response · Authors · 2021-11-18
> **Response to Reviewer 4pht**
>
> Thank you for reviewing our paper and acknowledging the novelty and usefulness of this work.
>
> We would like to address your concerns as follows.
>
> ### Single-parameter
>
> There are two dimensions that we need to consider while modeling instance-dependent noise: **group-level vs. instance-level information**, and **full vs. approximated transition**.
> In CCN, we only considered a full transition matrix to extract group-level information.
> It was shown that a full transition matrix for each instance may end up with high approximation error (Adaptation).
> Therefore, we went to the other end and studied if a single-scaler is enough to extract some instance-level noise information.
> It turns out to be quite useful in applications such as mislabeled instance detection.
> Another benefit of using a single parameter is that it is easy to sort and compare training examples, as shown in Figures 5 and 6.
>
> Indeed, based on those confidence parameters after training, we only know how likely an instance is mislabeled, but not which class it should be.
> If such information is crucial for the task, we may extend this method and use maybe $2$ or $K$ parameters for each instance.
> The design of such approximations is left for future work.
> Note that it may introduce more estimation error and the result may be less reliable.
>
> Another direction is to combine ICE with Forward (a constant transition matrix for all instances) because ICE only affects the class-posterior and is usually compatible with other training methods.
> In this way, class-wise noise information is captured in the transition matrix and instance-level noise information is captured in the confidence parameters.
> Such information may be useful, for example, for designing active learning query strategies.
>
> In conclusion, single-parameter ICE is already useful and effective enough for some applications, and it can be extended or combined with other methods according to the task.
>
> ### Intuition
>
> $g$ is a mapping from an instance to a trainable parameter.
> How this parameter is used is defined via $h$.
>
> To assign a trainable parameter to each instance, we have several choices.
> For example, we can use a neural network and train it in an end-to-end fashion.
> However, because two instances that are close in the feature space may have very different confidence parameters, a neural network may output too smoothed values.
> Further, the model is designed in a way that we do not need the generalization ability for this part (partially due to the argmax-preserving requirement for $h$).
> Then, it is possible to assign a trainable parameter to each instance, hence the "instance embedding".
>
> For further discussion, please also refer to Appendix D.
>
> ### Baselines
>
> We agree that a few methods for IDN have appeared in the literature.
> Some baselines here are designed for CCN, such as Forward and Dual-T.
> Some robust losses do not assume a noise model (although theoretical analysis may do), such as Bootstrapping and GCE.
> Adaptation is a baseline for IDN where a full transition matrix for each instance is estimated.
> Further, we tested not only label noise methods, but also an abstention/rejection method DAC.
>
> To improve the credibility, we have added [2] PLC (Zhang et al. (2021a)) to Table 1 in the revised paper.
> We found that PLC performs well, especially in SVHN (PLC $86.23\\%$ vs. ICE-POW $81.82\\%$).
> We hypothesize that it is due to images with multiple digits in SVHN (see also Figures 11 and 12 in Appendix E).
> Meanwhile, ICE still performs comparably to PLC on other datasets.
>
> We have not compared with [1] (Part-dependent Label Noise) and [5] (Confidence Score) because they either require domain-specific knowledge or extra supervision, which was discussed in the introduction.
> To avoid such limits is one of our motivations to do this research.
> We would like to point out that this method was adapted from image classification to text classification smoothly, whose result was quite surprising to us (e.g., Table 3).
> It might not be possible if we have some image-specific assumptions.
>
> Lastly, we would like to emphasize that we did not make the claim that ICE is a complete and optimal learning system for IDN that achieves state-of-the-art performance.
> We tried to keep it relatively simple, easy to implement, and compatible with other training methods.
> In this work, we verified its effectiveness and training dynamics (e.g., Figure 4 ridgeline plots), and found some interesting applications in mislabeled instance detection (Figure 5, Table 3, and more in Appendix E) on both image and text datasets.
> We also look forward to its potential applications in data cleansing, learning with rejection, or active learning.
> This is why we would like to share this idea with the community.
>
> ### Datasets
>
> Thank you for your nice suggestion!
> We would like to consider these real-world datasets in our future work.

---

> > ### Comment · Reviewer_4pht · 2021-12-01
> > **My unsolved concerns**
> >
> > Thanks authors for the detailed explanations about the use of single scalar as well as the intuition of $g$. My unsolved concern is that:
> >
> > In my first point **A single-scalar V.S. $K\times K$ noise transition matrix**, I agree with the authors that this can be beneficial in mislabeled instance detection. However, I am still not clear whether a single-scalar will neglect too much crucial information when learning with label noise, either from a theoretical aspect or an empirical view. Especially for the experiments:
> >
> > - In Appendix E (Table 5), the performance of ICE-POW on Clothing1M is not very promising (by referring to DMI);
> > - Lack of comparisons with other methods that targeting at IDN in small-scale datasets.
> >
> > These two experiment issues (the second one also mentioned by other reviewers) make me not fully convinced of the effectiveness of ICE when learning with label noise. And my first point is not well addressed.

---

### Official Review · Reviewer_ZuLD · 2021-11-02

**Correctness:** 2
**Technical Novelty And Significance:** 2
**Empirical Novelty And Significance:** 2
**Recommendation:** 3
**Confidence:** 4

**Main Review:**

Strength:
The suggested method captures instance-specific noise information and thereby improves classification performance when compared to previous methods based on the class-conditional noise (CCN) assumption.


Concerns:
1.	For an instance X with label Y=i, how do you define $T_{kj}(x)$, where $k \neq i$.
2.	At the bottom of page 2, matrix-valued function $T$ is introduced. Is it different from the transition matrix $T(x)$?
3.	Why do you need $K\times K$ parameters to model the transition relation for each instance X while the $p$ and $q$ you defined are both $\in \Delta^{K-1}$?
4.	The previous works employed some assumptions or are based on domain-specific knowledge but they fully model the IDN. In your method, the X is replaced by one postulated variable C learned by embedding an index parameter. Is this assumption strong?
5.	The authors claim that most columns of the IDN transition matrix have only limited influence on the class-posterior estimation and the transition function should be argmax-preserving function. Does it mean this method can only deal with label noise with a low noise rate?
6.	$h(\boldsymbol{p} ; 0)=\boldsymbol{u}$, where $\boldsymbol{u} \in \Delta^{K-1}$. Does it mean $u$ is uniformly sampled front the rest $K-1$ classes expect the true class? Why is that?
7.	In Eq. (8), what is $u_i$? Is it the $i-th$ element of $u$, then what is $u$?
8.	Eq. (9) needs to be elaborated.
9.	The choices of $h$ and $g$ seem come from nowhere and need more justification. Why can this method work?
10.	Important baselines are missing. All the IDN methods the authors mentioned in the introduction part were not considered as baselines.
11.	Some grammatical issues should be fixed.

**Summary Of The Paper:**

This paper presented an instance-confidence embedding model (ICE), which is a variational approximation of the instance-dependent noise (IDN). Given the observation that most columns of the IDN transition matrix have only limited influence on the class-posterior estimation, this paper uses a single-scalar confidence parameter to model the noise and uses a simpler transformation to transfer clean class posterior to noisy class posterior. The suggested method captures instance-specific noise information and thereby improves classification performance when compared to previous methods based on the class-conditional noise (CCN) assumption.

**Summary Of The Review:**

Overall, the proposed method is too heuristic without enough justification. The experiments are not sufficient.

---

> ### Author Response · Authors · 2021-11-18
> **Response to Reviewer ZuLD**
>
> Thank you for reviewing our paper.
> We would like to address your concerns as follows.
>
> 1. As defined in Section 2.2 (Page 2), $\mathbf{T}_{ij}(x) := p(\tilde{Y}=j | Y=i, X=x)$ for $i, j \in \\{1, \dots, K\\}$.
> 2. A transition matrix that depends on $x$ is viewed as a matrix-valued function $\mathbf{T}: \mathcal{X} \to [0, 1]^{K \times K}$.
> 3. A transition matrix with $K$ rows and $K$ columns contains $K \times K$ parameters. A $(K-1)$-dimensional probability [simplex](https://en.wikipedia.org/wiki/Simplex) $\Delta^{K-1}$ contains $K$-dimensional vectors, which is defined as $\\{\mathbf{p} \in \mathbb{R}^K: p_1 + \dots + p_K = 1, p_i \geq 0, i = 1, \dots, K\\}$. For example, for $3$-class classification, parameters are in $\Delta^2$ (Figure 2). By the way, the degrees of freedom are $(K-1) \times K$ because each row of the transition matrix sums to $1$.
> 4. This is the design of the approximation, not an assumption. Among previous works, only [Goldberger & Ben-Reuven, 2017] fully modeled IDN but the model is not identifiable and the estimation error could be high. Other existing methods either (1) also approximate IDN, (2) need more assumptions on the distribution, or (3) need extra supervision.
> 5. Most columns have limited influence because we assume that the clean labels should be "clean", i.e., close to a vertex of the simplex. It still holds even if the noise rate is high.
> 6. See (3) the definition of simplex, and the definition of $\mathbf{u}$ in Section 3.2 (Page 4). It is a uniform probability vector $u_i=\frac1K$, not "uniformly sampled". For example, $\mathbf{u} = [1/3, 1/3, 1/3] \in \Delta^2$.
> 7. See (6) and also Appendix C.
> 8. See also Section 4 temperature scaling.
> 9. Section 3.2 explains the design of $h$ and Section 3.3 explains the design of $g$. For why this method works, see also Appendices B and D.
> 10. Adaptation (fully IDN), Forward (CCN approximation), and DAC (abstention) are all mentioned in the introduction. CCE, Bootstrapping (regularization), Dual-T (CCN approximation), and GCE (robust loss) are discussed in Section 4 related work. Other methods either only focus on binary classification under strong assumptions (Menon et al., 2018; Cheng
> et al., 2020), are based on domain-specific knowledge (Xia et al., 2020), or need extra supervision (Berthon et al., 2021), which made them not applicable (at least not easily) to this problem. This is one of the motivations that we present this approach.
> 11. We would be grateful if you could point out which grammatical issues so we can improve our manuscript. Thank you.

---

> > ### Comment · Reviewer_ZuLD · 2021-11-30
> > **Response**
> >
> > Thanks for your efforts. Below are some further questions:
> >
> > 1. My question is that for an instance $x$ with label $y=i$, how do you define $T_{kj}(x)$, where $k \neq i$ rather than $T_{ij}(x)$. Let me make it more clear: $T_{ij}(x) = \frac{P(\tilde{y} = j, y=k, x)}{P(y=k, x)}$, will the denominator be zero when $x$ is an anchor point?
> >
> > 2. The authors mentioned that $\boldsymbol{T}(x)$ serves as a linear mapping from $\boldsymbol{p}$ to $\boldsymbol{q}$ (Eq. (3)), and also in implementation, $h$ is employed to model this transformation. If $h$ is required to be argmax-preserving, will $\boldsymbol{T}(x)$ also be argmax-preserving?
> >
> > 3. All the IDN methods the authors mentioned in the introduction part are Menon et al., 2018; Cheng et al., 2020, Xia et al., 2020, and Berthon et al., 2021, which are not considered as baselines. In Xia et al., 2020 and Berthon et al., 2021, experiments were conducted on benchmark datasets SVHN, F-MNIST, and CIFAR10. I don't understand why their methods can not be compared on those datasets.
> >
> > 4. Real-world dataset Clothing1M is a benchmark for IDN methods. Xia et al., 2020 and Berthon et al., 2021 both evaluated their methods on it. To show the effectiveness of your method, it will be more convincing if your method can perform well on it.

---

> > > ### Author Response · Authors · 2021-11-30
> > > **Response to Reviewer ZuLD**
> > >
> > > Thank you for the additional questions.
> > >
> > > 1. Please see [regular conditional probability](https://en.wikipedia.org/wiki/Regular_conditional_probability) in [Markov kernel](https://en.wikipedia.org/wiki/Markov_kernel#Regular_conditional_distribution).
> > > 2. $\mathbf{T}(x)$ may not be argmax-preserving, which results in approximation error.
> > > 3. [Menon et al., 2018] and [Cheng et al., 2020] are only for **binary classification**, and our task is multiclass classification. [Xia et al., 2020] need to decompose data into **non-negative factor matrices**. [Berthon et al., 2021] requires that each instance-label pair is equipped with a **confidence score**. The experiments may be based on the same dataset but they used different semi-synthetic data.
> > > 4. Please see Appendix E.

---

### Official Review · Reviewer_cgg6 · 2021-11-02

**Correctness:** 3
**Technical Novelty And Significance:** 3
**Empirical Novelty And Significance:** 3
**Recommendation:** 6
**Confidence:** 4

**Main Review:**

Overall this work has enough novel and significant contributions, but some issues with the experiments weaken the work. The single-parameter noise model and the use of ICE to learn with IDN are novel to my best knowledge, though aspects of these ideas exist in other literature. The paper does a good job describing the proposed method, but the experiment section lacks some important details.

---
Strengths:

- The single-parameter noise model proposed in this work is somewhat novel, and the idea is relatively simple (which is a good thing). This simplicity allows easier estimation (smaller estimation error), though it may have larger approximation error, compared with methods that model general noise transition function $T(x)$. But it indeed has a good balance of both approximation error and estimation error, in the middle of CCN (where $T(x)$ is a constant matrix) and general $T(x)$. This noise model is well-motivated.

- The use of ICE to overcome the issue where $T(x)$ could vary significantly for two adjacent instances is interesting and seems beneficial.

---
Weaknesses:

1. Lack of theoretical results. Variational approximation in this paper is only used to justify that the proposed single-parameter noise model leads to a valid lower bound. In fact, one can use any noise model and still get a valid lower bound using the same justification. The paper does not provide any theoretical analysis of the gap. Also, the connection with CCN is not clear. It looks like the proposed noise model does not include CCN as a special case.

2. In the experiments, it is not clear how that clean labels and noise are generated. The authors only mentioned that they followed a similar way used in Zhang et al. (2021a). However, the exact details are missing in the paper and the appendix. I am kindly asking the authors to provide this information. Some specific questions:

- In Zhang et al. Section 4, they first trained a model to approximate the posterior class probability $\eta(x)$ using the whole data set. Then they sampled $y_x \sim \eta(x)$ for each instance $x$. Instead of using the original labels, they used the sampled labels as clean labels. Was this paper following the same approach?

- In Zhang et al. Section 4, several types of noise generation are considered. Which type did this paper use? If for different data sets different types were used, please list them accordingly.

- In Table 1 of this paper, how was the overall noise rate controlled? If this paper followed Zhang et al. by multiplying the noise function by some constant factor, please specify the constants. Please also list the controlled noise rate for each data set.

3. This paper did not compare with the method in Zhang et al. (2021a) without any justification, even though the paper used Zhang et al.'s setting for noise generation. Please add the method in Zhang et al. to Table 1, or provide explanation why you chose not to compare with Zhang et al.

4. In Section 3.2, one important assumption is that the noisy posterior class probability ($q$) and the clean posterior class probability ($p$) share the same argmax. This is quite a strong assumption, because effectively, with this assumption, one can train directly on the noisy examples using standard methods and predict correctly (in the Bayes optimal sense). In fact, such assumption is not necessary for good predictions. For example, in CCN setting, if the noise transition matrix is invertible, one can train a good classifier using noisy examples and predict well w.r.t. the clean distribution. In Zhang et al., no such assumption was made.  A related question: how is this assumption enforced in the noise generation for the experiments?


**Summary Of The Paper:**

This paper studies instance-dependent noise (IDN) problems. It proposes a single-parameter (confidence) model for the noise generating process. Moreover, *instance-confidence embedding* (ICE) method is employed in the training process. The experimental results on various image and text classification tasks confirm the effectiveness of the proposed method.

**Summary Of The Review:**

Overall this work has enough novel and significant contributions (single-parameter noise model and ICE). But it has a strong assumption (argmax preserving) and lacks theoretical justifications. More importantly, the experiment section lacks detailed descriptions and did not compare with Zhang et al. (2021a). These shortcomings make me skeptical about the effectiveness of the proposed method. Therefore, I vote for a weak reject.

---

After rebuttal:

I raised my score to 6. See my reply below for more details.

---

> ### Author Response · Authors · 2021-11-18
> **Response to cgg6**
>
> Thank you for your time and efforts in reviewing our paper and checking the experimental details, which would help us to improve its clarity and soundness.
>
> We would like to address your concerns and clarify the experimental setting as follows.
>
> ### Theoretical analysis
>
> We agree with the reviewer that more theoretical results would improve the credibility of this method.
> Here, we mainly focused on the methodology and its potential applications in mislabeled instance detection and label issue diagnosis.
> If we want stronger theoretical guarantees, we may need to **restrict the problem and impose more assumptions** on the distribution or transition matrix, which is somehow against the purpose of this research.
>
> As for the variational approximation, indeed we can obtain the same lower bound for other models.
> However, **we cannot use an arbitrary approximation**.
> We explained our model design that we only choose argmax-preserving functions as $h$ in Section 3.2.
> Because of this design, the decision based on $h(f(x), g(x))$ during training is the same as the decision based on only $f(x)$ during test when we do not need (and do not have) the confidence parameters.
> Also because of this restriction, we may theoretically characterize the gap between the approximation and the actual IDN.
> However, it may not influence how practitioners can use this method so we omitted this part for the sake of simplicity.
>
> As for its connection with CCN, yes, **it does not include CCN as a special case**.
> If the noise is actually instance-independent, then the linear interpolation approximation may capture the noise transition matrix when the noise is symmetric (by assigning $(1 - \text{noise rate})$ to all confidence parameters), otherwise, both approximations can never be exactly the same as the transition matrix, which causes approximation error.
> However, due to the argmax-preserving restriction, it does not influence the final decision.
> If confidence calibration is required, post hoc calibration methods such as temperature scaling and Dirichlet calibration can be applied.
>
> Nevertheless, theoretical justification is indeed one of our future directions for this research.
>
> ### Experimental setting
>
> We would like to clarify the experimental settings. More specifically, we used a **hybrid noise with $20\\%$ Type I (Zhang et al. (2021a)) + $40\\%$ Uniform**. The overall noise rate is around $52\\%$. We have updated the statement in the revised paper to make it clearer.
>
> For each question:
>
> - Yes, we followed the same procedure.
> - Type I, as the procedure used [here](https://github.com/pxiangwu/PLC/blob/master/utils.py#L149).
> - We mean that the overall noise rate varies slightly due to overlay. We have updated the statement.
>
> ### Baseline
>
> We have added PLC (Zhang et al. (2021a)) to Table 1 in the revised paper.
> We found that PLC performs well in this noise setting, especially in SVHN (PLC $86.23\\%$ vs. ICE-POW $81.82\\%$).
> We hypothesize that it is due to images with multiple digits in SVHN (see also Figures 11 and 12 in Appendix E).
> Meanwhile, ICE still performs comparably to PLC on other datasets.
>
> ### Argmax-preserving
>
> First of all, we did not make a **data assumption** that noisy class-posterior and clean class-posterior have the same top-1 index, which excludes noises other than symmetric noise or restricts clean class-posterior to be totally deterministic.
> Otherwise, as the reviewer said, we can train with noisy labels directly (given that we have infinite training examples).
>
> Restricting the approximation $h$ to be argmax-preserving is a **model design**, which indeed introduces approximation error.
> However, we found that such an approximation is acceptable and leads to a practically useful method.
> Further, since we only have access to an (assumably) i.i.d. sample, not the distribution itself, we do not (and probably cannot) enforce this requirement in the noise generation.
> We generated synthetic label noises as described above.
>
> ### Contribution
>
> Lastly, we would like to emphasize that we did not make the claim that ICE is a complete and optimal learning system for IDN that achieves state-of-the-art performance.
> We tried to keep it relatively simple, easy to implement, and compatible with other training methods.
> For example, to capture both inter-class label noise and instance-dependent labeling error, it is possible to use ICE along with Forward (a constant transition matrix for all instances).
> In this work, we verified the effectiveness and training dynamics (e.g., Figure 4 ridgeline plots) of ICE and found some interesting applications in mislabeled instance detection (Figure 5, Table 3, and more in Appendix E) on both image and text datasets.
> We also look forward to its potential applications in data cleansing, learning with rejection, or active learning.
> This is why we would like to share this idea with the community.

---

> > ### Comment · Reviewer_cgg6 · 2021-11-29
> > **Raise my score to 6.**
> >
> > Thank the authors for the detailed rebuttal. I have also read the other reviews and their responses. Overall I think this work has enough novel and significant contributions (single-parameter noise model and ICE) with good and interesting motivations, though it lacks theoretical justifications. I raised my score to 6 because the authors added some more details and the comparison with PLC from Zhang et al. (2021), which partially relieved my doubts. My main concern is still about the experiments (as also pointed out by the other reviewers).
> >
> > - The baseline choice is a bit problematic. The comparisons with other IDN methods should have been included initially. At least Zhang et al. (2021) should have been included. This shows some issues with the original experimental design.
> >
> > - The model is build on argmax-preserving assumption (or called "model design" by the authors), so it is reasonable to see better performance when this assumption is satisfied (in other words, when the data distribution satisfies the model design). With the current noise generation process, the percentage of instances whose noisy posterior class probability and clean posterior class probability share the same argmax, is not clearly known. Based on my understanding of the noise generation process used in the experiments, it may very well be the case that for some data sets (for example, MNIST and FMNIST) that percentage might be high, because CCE performed quite well even though it was trained on noisy data. It would have helped better understand the effectiveness of the proposed method if two families of noise generation processes were considered: 1. percentage of instances whose noisy posterior class probability and clean posterior class probability share the same argmax, is high; 2. percentage of instances whose noisy posterior class probability and clean posterior class probability share the same argmax, is low. Such percentage can be reported easily with the current experimental design because the clean and noisy posterior class probabilities are known (see the first bullet under Weakness 2 in my review), but it cannot be inferred from the overall noise rate reported. To summarize, with only one noise generation considered in the experiments, it is hard to fully evaluate the effectiveness of the proposed method.
> >
> >  To sum up, the idea proposed in this paper is relatively simple and interesting. It can potentially be applied to other learning task as well. The experiment results show its promising performance. However, the lack of different noise generation processes and the comparison with other IDN methods makes it hard to fully evaluate the effectiveness of the proposed method. That's why I decided to give a score of 6 instead of 8.

---

> > > ### Author Response · Authors · 2021-11-30
> > > **Thank you for the detailed feedback!**
> > >
> > > We thank Reviewer cgg6 for reviewing our paper and providing constructive suggestions! We would like to further clarify the argmax-preserving assumption/design and investigate the behavior of the proposed method under different noise generation processes in the revised paper. Your review helps us to improve this manuscript. Thank you!

---

### Decision · Program_Chairs · 2022-01-20

**Decision:**

Reject

**Comment:**

The paper proposes a new method for the problem of learning under instance-dependent noise (IDN). The idea is to construct a variational approximation to the ideal training objective, which involves learning a single scalar C(x) per instance. In turn, each such scalar is treated as an additional parameter to be learned by the network.

Reviewers generally found the basic idea of the proposal to be interesting and novel, with the response clarifying some initial questions on the design of the network to learn C(x). The paper is also well-written, and presents experiments on image and text classification benchmarks. Some concerns were however raised:

(1) _Limited theoretical justification_. There is limited formal analysis of when the proposed method can work well.

(2) _Lack of comparison to IDN baselines_. The original submission did not include any IDN baselines as comparison. The revision included results of the method of (Zhang et al., '21a), which is on-par or better than the proposed method; it seems that this baseline really ought to have been included in the original submission, but it is appreciated that these have been added. A related concern was the marginal gains over the GCE method on the CIFAR datasets.

(3) _Sufficiency of learning a single parameter_. The paper learns a single scalar per sample. Several reviewers were unsure on the sufficiency of this parameter to capture the underlying noise distribution.

For (1), the authors acknowledge theoretical analysis as an important future direction. This is perfectly reasonable, but does then require weighting more any issues with the the conceptual and empirical contributions of the paper.

For (2), the response clarified that most of these operate either in the binary setting, or require auxiliary information. This is a valid motivation for the present work; it would however be more compelling to include results in a binary setting, to better understand the strengths and weaknesses compared to existing proposals. The response also clarified the present method does not claim to improve upon state-of-the-art performance, but rather proposes a simple method which has additional applications (as shown in Appendix E). This is a reasonable claim; however, to my taste, there is insufficient discussion of the PLC method (Zhang et al., '21a), and what new conceptual information the present work offers.

For (3), the response argued that the present results already demonstrate the efficacy of using a single parameter, and that using multiple parameters can be studied in future work. One reviewer was not convinced of the efficacy being shown in some of the results in Appendix E. It could strengthen the work if there is an empirical analysis of when the single parameter assumption starts to break down; e.g., perhaps under increasing levels of CCN noise?

Overall, the paper has interesting ideas and some nice analyses. At the same time, there was clear scope for improvement in the original submission. This was partially addressed in the revision, but given that several domain experts retain reservations (particularly in regards to comparisons against prior IDN works), it is encouraged that the authors incorporate the above comments for a future version of the paper.